# Who Said Neural Networks Aren't Linear?

**Nimrod Berman**[*]
Ben-Gurion University
bermann@post.bgu.ac.il

**Assaf Hallak**[*]
NVIDIA
ahallak@nvidia.com

**Assaf Shocher**[*†]
Technion
assaf.sh@technion.ac.il

## Abstract

Neural networks are famously nonlinear. However, linearity is defined relative to a pair of vector spaces, $f : \mathcal{X} \to \mathcal{Y}$. Leveraging the algebraic concept of transport of structure, we propose a method to explicitly identify non-standard vector spaces where a neural network acts as a linear operator. When sandwiching a linear operator $A$ between two invertible neural networks, $f(x) = g_y^{-1}(Ag_x(x))$, the corresponding vector spaces $\mathcal{X}$ and $\mathcal{Y}$ are induced by newly defined addition and scaling actions derived from $g_x$ and $g_y$. We term this kind of architecture a Linearizer. This framework makes the entire arsenal of linear algebra, including SVD, pseudo-inverse, orthogonal projection and more, applicable to nonlinear mappings. Furthermore, we show that the composition of two Linearizers that share a neural network is also a Linearizer. We leverage this property and demonstrate that training diffusion models using our architecture makes the hundreds of sampling steps collapse into a single step. We further utilize our framework to enforce idempotency (i.e. $f(f(x)) = f(x)$) on networks leading to a globally projective generative model and to demonstrate modular style transfer.

## 1. Introduction

Linearity occupies a privileged position in mathematics, physics, and engineering. Linear systems admit a rich and elegant theory: they can be decomposed through eigenvalue and singular value analysis, inverted or pseudo-inverted with well-understood stability guarantees, and manipulated compositionally without loss of structure. These properties are not only aesthetically pleasing, they underpin the computational efficiency of countless algorithms in signal processing, control theory, and scientific computing. Crucially, repeated application of linear operators simplifies rather than complicates: iteration reduces to powers of eigenvalues, continuous

evolution is captured by the exponential of an operator, and composition preserves linearity (Strang, 2022).

In contrast, non-linear systems, while more expressive, often defy such a structure. Iterating nonlinear mappings can quickly lead to intractable dynamics; inversion may be ill-posed or undefined; and even simple compositional questions lack closed-form answers (Strogatz, 2024). Neural networks, the dominant modeling tool in modern machine learning, are famously nonlinear, placing their analysis and manipulation outside the reach of classical linear algebra (Hornik et al., 1989). As a result, tasks that are trivial in the linear setting, such as projecting onto a subspace, enforcing idempotency, or collapsing iterative procedures, become major challenges in the nonlinear regime that require engineered loss functions and optimization schemes. This motivates our approach: applying the principle of conjugation (change of coordinates) to deep neural networks to induce a latent space where the mapping becomes linear. If so, we gain access to linear methods without sacrificing nonlinear expressiveness.

In this paper, we propose a framework that precisely achieves this goal. By embedding linear operators between two invertible neural networks, we induce new vector space structures under which the overall mapping is linear. We call such architectures *Linearizers*. This is a constructive claim about a specific architecture: a Linearizer is exactly linear w.r.t. a pair of *learned* vector spaces. We do not claim that off-the-shelf networks (MLPs, transformers) are secretly linear. This perspective not only offers a new lens on neural networks, but also enables powerful applications: collapsing hundreds of diffusion sampling steps into one, enforcing structural properties such as idempotency, and more. In short, Linearizers provide a bridge between the expressive flexibility of nonlinear models and the analytical tractability of linear algebra.

## 2. Linearizer Framework

Linearity is not an absolute concept; it is a property defined relative to a pair of vector spaces, $f : \mathcal{X} \to \mathcal{Y}$. This motivates the use of **transport of structure** (Mac Lane & Birkhoff, 1999) to identify a pair of spaces $\mathcal{X}$ and $\mathcal{Y}$, for which a function that is nonlinear over standard Euclidean

---

[*]Equal contribution.

*Proceedings of the 43rd International Conference on Machine Learning*, Seoul, South Korea. PMLR 306, 2026. Copyright 2026 by the author(s).

space is, in fact, perfectly linear. Recall that a vector space is comprised of a set of vectors (e.g. $\mathbb{R}^N$), a field of scalars (e.g., $\mathbb{R}$), and two fundamental operations: vector addition ($+$) and scalar multiplication ($\cdot$). We propose inducing new **learnable** vector spaces by redefining operations, while keeping vectors and field unchanged.

## 2.1. Definitions

We introduce a formalism we term *Linearizer*, in which the relevant vector spaces are immediately identifiable by construction as they are isomorphic to the Euclidean space. The architecture we propose can be trained from scratch or by distillation from an existing model. Our approach is made practical by invertible neural networks (Rezende & Mohamed, 2015; Dinh et al., 2015; 2017). We build our model by wrapping a linear operator, a matrix $A$ (the *core*), between two such invertible networks, $g_x$ and $g_y$:

**Definition 2.1** (Linearizer). Let $\mathcal{X}, \mathcal{Y}$ be two spaces and $g_x : \mathcal{X} \to \mathcal{X}$, $g_y : \mathcal{Y} \to \mathcal{Y}$ be two corresponding invertible functions. Also, let $A : \mathcal{X} \to \mathcal{Y}$ be a linear operator (the *core*). Then we define the Linearizer $\mathbb{L}_{\{g_x, g_y, A\}}$ as the following function $f : \mathcal{X} \to \mathcal{Y}$:

$$f(x) = \mathbb{L}_{\{g_x, g_y, A\}}(x) = g_y^{-1}(Ag_x(x)) \qquad (1)$$

We refer to $A$ as the *core* of the Linearizer throughout the paper. $g_x$, $g_y$, and the core $A$ are *learned jointly* end-to-end.[1]

For this construction, we can define the corresponding vector spaces $\mathcal{X}$ and $\mathcal{Y}$, also shown in Figure 1:

**Definition 2.2** (Induced Vector Space Operations). Let $g : V \to V$ be an invertible function. We define a new set of operations, according to the **transport of structure**, $\oplus$ and $\odot$, for any two vectors $v_1, v_2 \in V$ and any scalar $a \in \mathbb{R}$:

$$v_1 \oplus_g v_2 := g^{-1}(g(v_1) + g(v_2)) \qquad (2)$$
$$a \odot_g v_1 := g^{-1}(a \cdot g(v_1)) \qquad (3)$$

## 2.2. Linearity

The input space $\mathcal{X}$ is defined by operations $(\oplus_x, \odot_x)$ induced by $g_x$, and the output space $\mathcal{Y}$ by $(\oplus_y, \odot_y)$ induced by $g_y$. This is a **vector-space isomorphism** which promises preservation of geometry.

**Proposition 2.3.** $(V, \oplus_g, \odot_g)$ *is a vector space over $\mathbb{R}$. This is a direct consequence of the isomorphism. However, we also provide full verification in* Appendix A.

---

[1]Specifically, in our implementation each of $g_x$ and $g_y$ uses **6** invertible blocks (Appendix H.1). The specific INN architecture – affine coupling layers with a tiny U-Net conditioner – is detailed in Appendix H.2 and Appendix H.4. The linear core $A$ is task-specific and its implementations are summarized in Appendix H.3.

**Proposition 2.4.** *The function $f(x)$ is a linear map from the vector space $\mathcal{X}$ to the vector space $\mathcal{Y}$.*

*Proof.* Linearity is preserved in transport of structure, but we can also verify:

$$f(a_1 \odot_x x_1 \oplus_x a_2 \odot_x x_2)$$
$$= g_y^{-1}(Ag_x(g_x^{-1}(a_1 g_x(x_1) + a_2 g_x(x_2))))$$
$$= g_y^{-1}(a_1 g_y(g_y^{-1}(Ag_x(x_1))) + a_2 g_y(g_y^{-1}(Ag_x(x_2))))$$
$$= a_1 \odot_y f(x_1) \oplus_y a_2 \odot_y f(x_2) \qquad \square$$

## 2.3. Intuition

The Linearizer can be understood through several lenses.

**Linear algebra analogy.** The Linearizer is analogous to an eigendecomposition or SVD. Just as a matrix is diagonal (i.e. simplest) in its eigenbasis, our function $f$ is a simple matrix multiplication in the "linear basis" defined by $g_x$ and $g_y$. This provides a natural coordinate system in which to analyze and manipulate the function. For example, repeated application of a Linearizer with a shared basis ($g_x = g_y = g$) is equivalent to taking a power of its core matrix:

$$\mathbb{L}_{\{g,g,A\}}^{\circ N}(x) = \underbrace{f(f(\cdots f(x)))}_{N \text{ times}} \qquad (4)$$
$$= g^{-1}(A^N \cdot g(x)) = \mathbb{L}_{\{g,g,A^N\}}(x)$$

**Geometric interpretation.** $g_x$ can be seen as a diffeomorphism. Equip the input with the pullback metric $g_x^* \langle \cdot, \cdot \rangle_{\mathbb{R}^N}$ induced by the Euclidean metric in latent space. With respect to this metric, straight lines in latent correspond to geodesics in input, and linear interpolation in the induced coordinates maps to smooth and semantically coherent curves in input space.

## 2.4. Properties

We review how basic properties of linear transforms are expressed by Linearizers. All these properties follow directly from the vector space isomorphism (transport of structure). Yet, we explicitly derive them separately here.

### 2.4.1. COMPOSITION

**Proposition 2.5.** *The composition of two Linearizer functions with compatible spaces $f_1 : \mathcal{X} \to \mathcal{Y}$ and $f_2 : \mathcal{Y} \to \mathcal{Z}$, is also a Linearizer.*
$$(f_2 \circ f_1)(x) = g_z^{-1}(A_2 g_y(g_y^{-1}(A_1 g_x(x))))$$
$$= g_z^{-1}((A_2 A_1) \cdot g_x(x)) \qquad (5)$$

### 2.4.2. INNER PRODUCT AND HILBERT SPACE

**Definition 2.6** (Induced Inner Product). Given an invertible map $g : V \to \mathbb{R}^N$, the induced inner product on $V$ is

$$\langle v_1, v_2 \rangle_g := \langle g(v_1), g(v_2) \rangle_{\mathbb{R}^N}, \qquad (6)$$

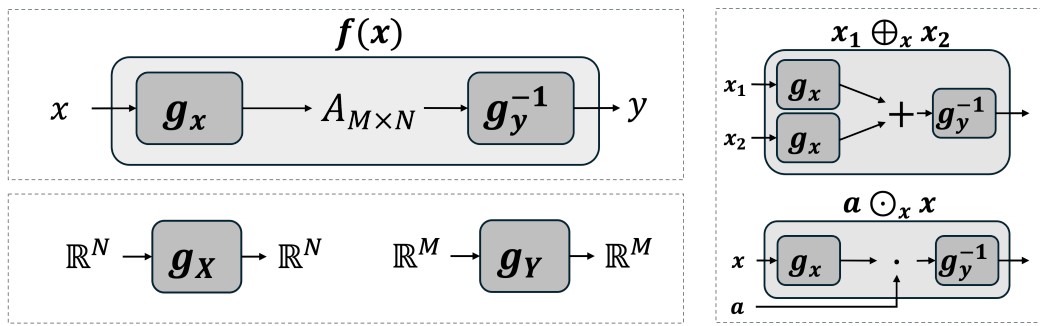

Figure 1. **Left.** The Linearizer structure (top) is a linear operation sandwiched between two invertible functions (bottom). **Right.** Vector addition and scalar multiplication define induced vector spaces for which $f$ is linear.

(a)                                                      (b)

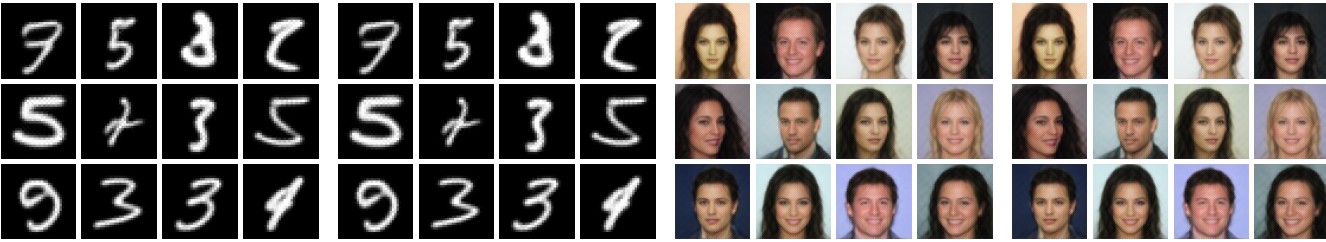

Figure 2. Comparison between multi-step and one-step flow matching. Panel labels: **(a)** multi-step Linear FM. **(b)** one-step Linear flow matching (FM).

where the right-hand side is the standard Euclidean dot product.

Equipped with this inner product, the induced vector spaces make *Hilbert spaces*.

**Proposition 2.7** (Induced spaces are Hilbert). Let $g : V \to \mathbb{R}^n$ be a smooth bijection and endow $V$ with the induced vector space operations $(\oplus_g, \odot_g)$ and inner product $\langle u, v \rangle_g := \langle g(u), g(v) \rangle_{\mathbb{R}^n}$. Then $(V, \langle \cdot, \cdot \rangle_g)$ is a Hilbert space. This is a characteristic of the transport of structure. For explicit proof *See Appendix C*.

### 2.4.3. TRANSPOSE

**Proposition 2.8** (Transpose). Let $f(x) = g_y^{-1}(Ag_x(x))$ be a Linearizer. Its transpose $f^\top : \mathcal{Y} \to \mathcal{X}$ with respect to the induced inner products is

$$f^\top(y) = g_x^{-1}\big(A^\top g_y(y)\big). \tag{7}$$

The transpose is always defined: in our finite-dimensional induced Hilbert spaces (Definition 2.6), the adjoint of a Linearizer exists by construction, since $A^\top$ is well-defined for any matrix $A$ and $g_x, g_y$ are invertible by assumption.

*Proof.* For all $x \in \mathcal{X}, y \in \mathcal{Y}$,

$$\langle f(x), y \rangle_{g_y} = \langle Ag_x(x), g_y(y) \rangle_{\mathbb{R}^N} = \langle g_x(x), A^\top g_y(y) \rangle_{\mathbb{R}^N} \tag{8}$$

$$= \langle x, g_x^{-1}(A^\top g_y(y)) \rangle_{g_x} = \langle x, f^\top(y) \rangle_{g_x}. \quad \square$$

### 2.4.4. SINGULAR VALUE DECOMPOSITION

**Proposition 2.9** (SVD of a Linearizer). Let $A = U\Sigma V^\top$ be the singular value decomposition of $A$, with $u_i$ denoting the $i$-th column of $U$ and $v_i$ the $i$-th column of $V$. Then the SVD of $f(x) = g_y^{-1}(Ag_x(x))$ is given by singular values $\Sigma$, input singular vectors $\tilde{v}_i = g_x^{-1}(v_i)$, and output singular vectors $\tilde{u}_i = g_y^{-1}(u_i)$.

Spectral properties are transferred in isomorphic change of coordinates. We show this explicitly in *Appendix B*.

### 2.4.5. PSEUDOINVERSE

Because the Linearizer is linear, its Moore–Penrose pseudoinverse is also a Linearizer:

**Lemma 2.10** (Moore–Penrose pseudoinverse of a Linearizer). *The Moore–Penrose pseudoinverse of $f$ with respect to the induced inner products is*

$$f^\dagger(y) = g_x^{-1}(A^\dagger g_y(y)). \tag{9}$$

*Proof.* We verify the four Penrose equations (Penrose, 1955) in Appendix D. $\quad \square$

**Regularization.** Because $A$ is an explicit linear operator, any linear-operator regularizer (Tikhonov $A_\lambda^+ = (A^\top A + \lambda I)^{-1}A^\top$, truncated SVD, spectral filtering, ridge) transfers to $f$ through $A$, yielding a Linearizer $f_\lambda^\dagger(y) = g_x^{-1}(A_\lambda^+ g_y(y))$ with the same $g_x, g_y$.

## 3. Theoretical Analysis

Before demonstrating applications, we address two fundamental theoretical questions: the expressive capacity of the Linearizer and the functional necessity of the core matrix.

### 3.1. Expressiveness

A natural concern is whether enforcing a linear bottleneck restricts the model's capacity. Strictly speaking, the architecture does impose topological constraints on the global function $f$. For example, the kernel (null space) of $f$ is determined entirely by the kernel of $A$. Because the kernel is necessarily a linear subspace, it cannot consist of an arbitrary finite set of points: a Linearizer cannot, for example, map exactly three distinct points to zero while mapping a fourth nearby point to non-zero, as this would violate the subspace property of the kernel in the latent space. A second example: in 1D ($\mathcal{X} = \mathcal{Y} = \mathbb{R}$), $A$ is a scalar and $g_x, g_y$ are monotone bijections, so $f$ is necessarily monotone – ruling out e.g. $x \mapsto x^2$. These constraints relax in higher dimensions; Theorem I.5 shows the Linearizer fits any finite dataset.

Despite this topological constraint, **we prove that the Linearizer is expressive enough to fit any finite training set.** The argument is two-step: ideal-diffeomorphism $g_x, g_y$ fit any finite dataset *exactly* (Lemma H.4), and INN realizations approximate this to arbitrary precision (Theorem H.5) – matching the practical guarantee the universal approximation theorem gives for MLPs. Theorem I.5 makes this precise: for any finite dataset $\{(x_i, y_i)\}_{i=1}^N$ and $\varepsilon > 0$, there exist INNs $g_x, g_y$ and a linear $A$ with $\sup_i \|f(x_i) - y_i\| < \varepsilon$.

How can the model be topologically constrained yet fit any dataset? The resolution lies in the degrees of freedom available outside the data support. For example, the required infinite null-space can lie outside of the data. This logic also applies to infinite continuous manifolds of lower dimension than the space, allowing generalization on high-dimensional data, such as images or text embeddings that often reside on a low-dimensional manifold

### 3.2. On the Role of the Core Operator A

Given the power of $g$, what is the role of the matrix $A$? We analyze this in two regimes.

**The General Regime ($g_x \neq g_y$).** Let the SVD of $A$ be $U\Sigma V^T$. We could define new invertible networks $g'_y = g_y \circ U$ and $g'_x = V^T \circ g_x$. The function would then be $f(x) = (g'_y)^{-1}(\Sigma g'_x(x))$. If we further allow the networks to absorb the scaling defined by the singular values in $\Sigma$, then the core operator could be reduced to a diagonal matrix of zeros and ones (note that this does not imply idempotency unless $g_x = g_y$). In this sense, the fundamental role of $A$ is

simply to define the rank of the function $f$.

**The Constrained Regime ($g_x = g_y = g$).** In applications like Flow Matching or Idempotency, the input and output spaces are identical so $g_x = g_y$. Here, the two networks are not independent. To preserve the structure $f(x) = g^{-1}(Ag(x))$, any transformation absorbed by $g$ must be inverted by $g^{-1}$. This rules out $g$ absorbing SVD elements $U, V$ as they are not necessarily mutual inverses. For diagonalizable matrices however, eigen-decomposition can be applied. Then, $A = Q\Lambda Q^{-1}$, where one could define $g' = Q^{-1} \circ g$ and $f(x) = (g')^{-1}(\Lambda g'(x))$. Here, $A$ determines the function's eigenvalues—a stronger role than just its rank—but this is still only the *spectrum* of $A$. In the near-isometric regime of our Idempotency application (Section 4.3), the full SVD becomes reparametrization-invariant, so singular values *and* vectors are intrinsic to $f$ (Appendix E).

**Practical Necessity: Functional Diversity.** The reduction above is only valid when learning a *single static* function: in that case a diagonal $A$ does suffice, since the orthogonal SVD factors can be absorbed into $g_x, g_y$. However, our applications typically utilize *families* of Linearizers that share the same $g_x, g_y$ with different $A$ matrices. For example, our flow-matching application employs a different $A$ for each timestep. The expressiveness inside such a family is dictated by the expressiveness of $A$, which would be significantly reduced if it were diagonal; in such families the full operator structure of $A$ – not merely its rank or spectrum – is what carries the modeling capacity.

## 4. Applications

Having established the theoretical foundations, we now present proof-of-concept applications that highlight advantages of our framework, leaving scaling and engineering refinements for future work.

### 4.1. One-Step Flow Matching

Flow matching (FM) (Lipman et al., 2023; Song et al., 2021b) trains a network to predict the velocity field that transports noise samples to data points. In our setting, diffusion models (Ho et al., 2020) and flow matching (Lipman et al., 2023) are, from an algorithmic standpoint, equivalent under standard assumptions (Gao et al., 2024); hence we use the two frameworks interchangeably. Traditionally, this velocity must be integrated over time with many small steps, making generation slow. We train flow matching using our Linearizer as the backbone, with a single vector space ($g_x = g_y = g$). The key property that helps us achieve one-step generation is the closure of linear operators. Applying the model sequentially with some simple linear operations in between steps is a chain of linear operations. This chain

$x_1$ $x_2$

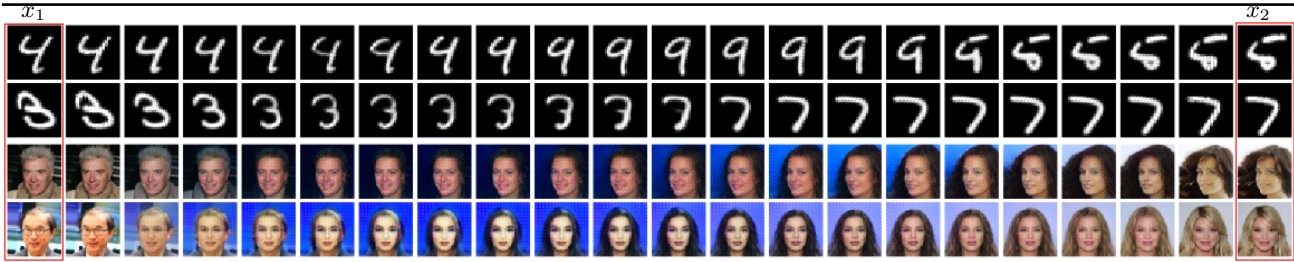

*Figure 3.* **One-step inversion examples:** Left and right (in red): original (not generated) data $x_1$ and $x_2$. Intermediate images obtained by latent interpolation. See Appendix F.2 for more and higher-resolution results.

can be collapsed to a single linear operator so that the entire trajectory is realized in a single step.

**Training.** Training is just as standard FM, only using the Linearizer model and the induced spaces. We define the forward diffusion process:

$$x_t = (1-t) \odot x_0 \oplus t \odot x_1 = g^{-1}\big((1-t)\, g(x_0) + t\, g(x_1)\big). \quad (10)$$

This is a straight line in the $g$-space, mapped back to a curve in the data space. The target velocity is

$$v = x_1 \ominus x_0 = g^{-1}\big(g(x_1) - g(x_0)\big). \quad (11)$$

We parameterize a time-dependent Linearizer

$$f(x_t, t) = g^{-1}(A_t g(x_t)), \quad (12)$$

and train it to predict $v$. The loss is

$$\begin{aligned}\mathcal{L} &= \mathbb{E}_{x_0, x_1, t} \left\| v \ominus f(x_t, t) \right\|^2 \\ &= \mathbb{E}_{x_0, x_1, t} \left\| g^{-1}\big(g(x_1) - g(x_0) - A_t g(x_t)\big) \right\|^2\end{aligned} \quad (13)$$

This objective ensures that the learned operators $A_t$ approximate the true velocity of the flow within the latent space of the Linearizer.

**Sampling as a collapsed operator.** In inference time, standard practice is to discretize the ODE with many steps. In the induced space, an Euler update reads

$$x_{t+\Delta t} = x_t \oplus \big(\Delta t \odot f(x_t, t)\big), \quad (14)$$

which expands to

$$g(x_{t+\Delta t}) = (I + \Delta t\, A_t)\, g(x_t). \quad (15)$$

Iterating this $N$ times produces

$$g(\hat{x}_1) \approx \underbrace{\left[\prod_{i=0}^{N-1} (I + \Delta t\, A_{t_i})\right]}_{:=B} g(x_0). \quad (16)$$

The entire product can be collected into a single operator $B$, resulting in

$$\hat{x}_1 = g^{-1}(Bg(x_0)). \quad (17)$$

Thus, what was originally a long sequence of updates collapses into a single multiplication. In practice, higher-order integration such as Runge–Kutta (Runge, 1895; Kutta, 1901) may be used to calculate $B$ more accurately (see Appendix G). $B$ is calculated only once, after training. Then generation requires only a single feedforward activation of our model on noise, regardless of the number of steps discretizing the ODE.

**Geometric intuition.** $g$ acts as a diffeomorphism: in its latent space, the path between $x_0$ and $x_1$ is a straight line, and the velocity is constant. When mapped back to the data space via $g^{-1}$, this straight line becomes a curved trajectory. The Linearizer exploits this change of coordinates so that, in the right space, the velocity field is trivial, and the expensive integration disappears.

**Implementation.** The operator $A$ is produced by an MLP whose input is $t$. To avoid huge matrix multiplications we build $A$ as a low-rank matrix, by multiplying two rectangular matrices. $g$ does not take $t$ as input. Full implementation details are provided in App. H.

**One-step generation.** Figure 2 shows qualitative results on MNIST (LeCun et al., 1998) (32×32) and CelebA (Liu et al., 2015) (64×64). As discussed, our framework supports both multi-step and one-step sampling and yields effectively identical outputs. Visually, the samples are indistinguishable across datasets; quantitatively, the mean squared error between one-step and multi-step generations is $\mathbf{3.0 \times 10^{-4}}$, confirming their near-equivalence in both datasets. Additionally, we evaluate FID across different step counts and our one-step simulation (see Table 1 (a)). The FID from *100 full iterative steps* closely matches our *one-step* formulation, empirically validating the method's theoretical guarantee of one-step sampling. Moreover, simulating more steps (1000→1 vs. 100→1) improves FID by ∼8 points, highlighting the strength of the linear-operator formulation. Furthermore, we validate one-step vs. 100 step fidelity, showing high similarity as presented in Table 1(c). Finally, while our absolute FID is not yet competitive with state-of-the-art systems, our goal here is to demonstrate the theory in practice rather than to exhaustively engineer for peak performance; scaling left to future work.

**Isolating the role of linearity.** Two ablations on CelebA at matched parameter budget show that linearity of $A$ – not the raw capacity of $g$ – is what enables one-step sampling. **(i)** Replacing $A$ with a nonlinear MLP of comparable capacity raises FID to **157** (Linearizer: 127 at 100 steps, 131 one-step from 100→1); since closure under composition is what enables the one-step collapse, no nonlinear core can recover this guarantee at any capacity. **(ii)** Training $g$ directly as a one-step flow with $t$ as input and no $A$ yields FID **164**. Both alternatives match or exceed our capacity yet perform worse.

**Inversion and interpolation.** A fundamental limitation of flow models is that, in contrast to VAEs (Kingma & Welling, 2014), they lack a natural encoder: they cannot map data back into the prior (noise) space, and thus act only as decoders. As a result, inversion methods for diffusion (Dhariwal & Nichol, 2021; Song et al., 2021a; Huberman-Spiegelglas et al., 2024) have become an active research area. However, these techniques are approximate and often suffer from reconstruction errors, nonuniqueness, or computational overhead. Our framework connects diffusion-based models with encoding ability.

Because the Linearizer is linear, its Moore–Penrose pseudoinverse is itself a Linearizer (Lemma 2.10, Section 2). This property enables the exact encoding of data in the latent space. For example, given two data points $x_1, x_2$, we can encode them via $z^a = (1-a) f^\dagger(x_1) + a f^\dagger(x_2)$, and decode back by $\hat{x}^a = f(z^a)$. Figure 3 shows latent interpolation between two real (non-generated) images. Additionally, we evaluate reconstruction quality using two standard metrics: LPIPS (Zhang et al., 2018) and PSNR. Table 1(b) shows inversion-reconstruction consistency. Our results confirm high-quality information preservation.

### 4.2. Modular Style Transfer

Style transfer has been widely studied since (Gatys et al., 2016) proposed optimizing image pixels to match style and content statistics. (Johnson et al., 2016) introduced perceptual-loss training of feed-forward networks, making style transfer practical.

**Linearizer formulation.** We fix $g_x$ and $g_y$ and associate each style with a matrix $A_{\text{style}}$. The matrix $A_{\text{style}}$ is produced by a hypernetwork that takes a *style index* as input (rather than time, as in the one-step setting). Then

$$f_{\text{style}}(x) = g_y^{-1}(A_{\text{style}} g_x(x)). \tag{18}$$

This separates content representation from style, making styles modular and algebraically manipulable. In practice, we distill $A_{\text{style}}$ from a pretrained Johnson-style network.

**Style interpolation.** Given two trained style operators, $A_{\text{style}}^x$ and $A_{\text{style}}^y$, we form an interpolated operator $A_{\text{style}}^{(\alpha x + (1-\alpha)y)}$ that represents a linear interpolation between them. We evaluated $\alpha \in \{0, 0.35, 0.40, 0.45, 0.50, 0.55, 0.60, 0.65, 1\}$,

$$f_{\text{style}}^{(\alpha x + (1-\alpha)y)}(x) = g_y^{-1}\left(A_{\text{style}}^{(\alpha x + (1-\alpha)y)} g_x(x)\right),$$

with results shown in Figure 4. Rows 1–3 interpolate between *mosaic/candy*, *rain-princess/udnie*, and *candy/rain-princess*, respectively. See App. F for additional transfers.

### 4.3. Linear Idempotent Generative Network

Idempotency, $f(f(x)) = f(x)$, is a central concept in algebra and functional analysis. In machine learning, it has been used in Idempotent Generative Networks (IGNs) (Shocher et al., 2024), to create a projective generative model. It was also used for robust test time training (Durasov et al., 2025). Enforcing a network to be idempotent is tricky and is currently done by using sophisticated optimization methods such as (Jensen & Vicary, 2025). The result is only an approximately idempotent model over the training data. We demonstrate enforcing accurate idempotency through architecture using our Linearizer. The key observation is that idempotency is preserved between the inner matrix $A$ and the full function $f$.

**Lemma 4.1.** *The function $f$ is idempotent $\iff$ The matrix $A$ is idempotent.*

*Proof.* For all $x$,

$$
\begin{aligned}
A^2 = A &\iff g^{-1}(A^2 g(x)) = g^{-1}(A g(x)) \\
&\iff \{g^{-1} \circ A \circ \underbrace{(g \circ g^{-1})}_{\text{Id}} \circ A \circ g\}(x) \\
&= \{g^{-1} \circ A \circ g\}(x) \\
&\iff f(f(x)) = f(x) \\
&\quad \text{(by the definition of } f) \quad \square
\end{aligned}
$$

**Method.** Figure 5 left shows the method. Recall that idempotent (projection) matrices have eigenvalues that are either 0 or 1. We assume a diagonalizable projection matrix $A = Q\Lambda Q^{-1}$ where $\Lambda$ is a diagonal matrix with entries $\{0, 1\}$. The matrices $Q, Q^{-1}$ can be absorbed into $g, g^{-1}$ without loss of expressivity. So we can train a Linearizer $g^{-1}(\Lambda g(\cdot))$. In order to have $\Lambda$ that is binary yet differentiable, we use the *straight-through estimator* (STE) (Bengio et al., 2013), having underlying probabilities $P$ as parameters in $[0, 1]$ and then use $A = \text{round}(P) + P - P.\text{detach}()$ where detach means stopping gradients. We forward propagate rounded values (0 or 1) and back propagate continuous values. The STE introduces gradient bias (the surrogate is not the true derivative of rounding); the forward-pass projector is nonetheless exactly idempotent and we observe stable training in practice.

| Full Steps | | | One-steps | | | PSNR | LPIPS | | | PSNR | LPIPS |
|---|---|---|---|---|---|---|---|---|---|---|---|
| 1 | 10 | 100 | (100→1) | (1000→1) | | | | | | | |
| 405 | 152 | 127 | 131 | 123 | MNIST | 31.6 | .008 | | MNIST | 32.4 | .006 |
| | | | | | CelebA | 33.4 | .006 | | CelebA | 32.9 | .007 |



*(a)* FID comparison (CelebA).     *(b)* Inversion-reconstruction consistency.     *(c)* 100 vs. 1 step fidelity.

*Table 1.* Quantitative comparisons.



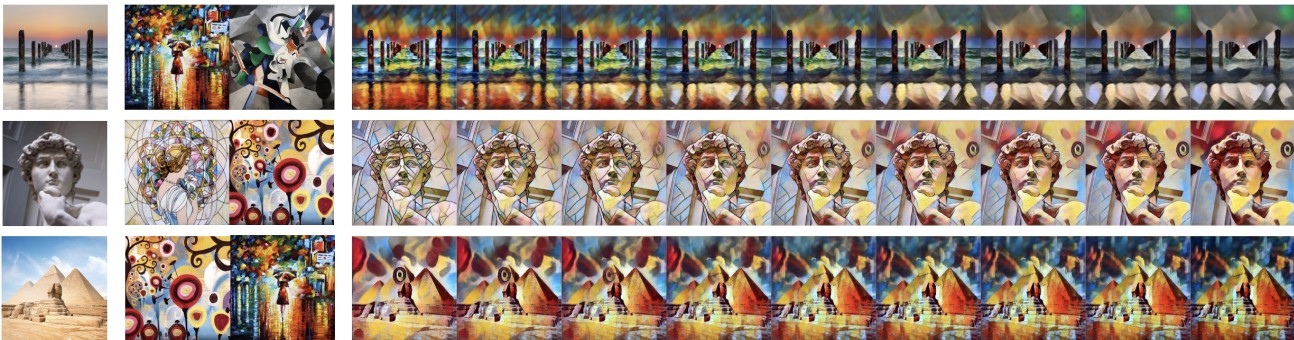

*Figure 4.* **Style transfer examples.** Left: original image. Middle: style transfer using the left-side and right-side style images. Right: interpolation between the two styles.

Because idempotency is now guaranteed by architecture (Lemma 4.1 above), we no longer need the explicit idempotency loss $\|f(f(x)) - f(x)\|^2$ used in (Shocher et al., 2024); only fixed-point and manifold-tightness terms are needed. We need the data to be fixed points with the tightest possible latent manifold. The losses are:

$$\mathcal{L}_{\text{rec}} = \|f(x) - x\|^2 \qquad (19)$$

$$\mathcal{L}_{\text{sparse}} = \frac{1}{N}\text{Rank}(A) = \frac{1}{N}\sum_i^N \lambda_i \qquad (20)$$

We further enforce regularization that encourages $g$ to preserve the norms:

$$\mathcal{L}_{\text{isometry}} = \left\| \|g(x) - g(0)\|^2 - \|x\|^2 \right\|_1 \qquad (21)$$

This nudges $g$ towards a near-isometry around the data, thus resisting mapping far-apart inputs to close outputs. It mitigates collapse and improves stability. We apply it with a small weight (0.001).

**From local to global projectors.** As Shocher et al. (2024) put it, *"We view this work as a first step towards a global projector."* In practice, IGNs achieve idempotency only around the training data: near the source distribution or near the target distribution they are trained to reproduce. If you venture further away, then the outputs degenerate into arbitrary artifacts not related to the data distribution. In contrast, our Linearizer is idempotent by architecture. It does not need to be trained to approximate projection, it is one. Figure 5 right shows various inputs – pure noise, structured patterns, and out-of-distribution shapes – projected by our model onto the digit manifold; see caption for evaluation

criteria. Interestingly, there is not even a notion of a separate *source distribution*: we never inject latent noise during training, and in a precise sense the entire ambient space serves as the source. This makes our construction a particularly unusual generative model and a natural next step toward the global projector envisioned in Shocher et al. (2024).

### 4.4. Additional Experiments

We provide more experiments of diverse types of data and tasks to empirically test the robustness of the Linearizer and its ability to learn various mappings.

**CelebFaces Attributes (CelebA).** We evaluate on the CelebA face-attribute classification task, which includes large pose variation, background clutter, diverse identities, and rich annotations (40 binary attributes such as *Male*, *Eyeglasses*, *Blond Hair*). After training, our Linearizer attains a test accuracy of **87.6%**; for reference, a ResNet-50 baseline reaches **86.1%**. These results suggest that the Linearizer handles moderately complex, real-world visual variability.

**Distillation of nonlinear disentanglement models.** In this experiment, we distill the encoder of a pretrained disentanglement model into a Linearizer (using pretrained checkpoints). We consider VAE, FactorVAE, and $\beta$-VAE on the **dSprites** dataset, which provides ground-truth factors of variation. We assess quality with standard metrics: **FactorVAE**, **SAP**, and **DCI** (disentanglement, completeness, informativeness). Across models, the Linearizer closely matches the original encoders (rows prefixed with *Linearizer-*) on these metrics (often slightly improving FactorVAE/SAP, e.g., BetaH: +0.10/+0.01), indicating that it can capture highly nonlinear transformations with no loss in disentanglement quality of the encoded representations. For the

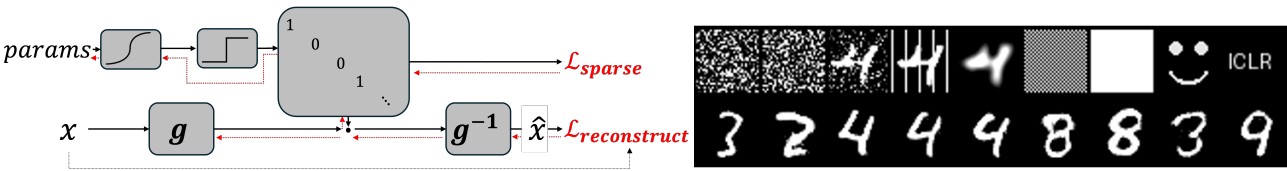

*Figure 5.* **Left: IGN training diagram.** Black solid arrows denote the forward pass; the red dashed arrow shows backpropagation. The parameter logits are passed through a sigmoid and then thresholded to form the binary projector $A$; during backprop, the straight-through estimator (STE) bypasses the threshold so gradients flow through the underlying probabilities. **Right: Projection results – successful global projection.** Each pair: arbitrary input (top: noise, patterns, OOD shapes) and its projection $f(x)$ onto the digit manifold (bottom). These are *successes*; criteria are (i) plausible digit, (ii) $f(f(x)) = f(x)$ (idempotent by architecture). Our Linear IGN makes a global projector that projects any input to the target distribution.

table with full results please see Appendix F.3.

**Weather Prediction.** To further explore the applicability of our framework, we conduct a small experiment in a new domain: regression of future steps. We use a standard weather-prediction benchmark where the model must predict the future given 96 past time steps. We compare our method with two well-known baselines tailored for regression, **Autoformer** (Wu et al., 2021) and **PatchTST** (Nie, 2022). Briefly, although our method is generic, it shows competitive results, surpassing the highly nonlinear Autoformer and being comparative (e.g., gaps of ∼0.008–0.034 MSE depending on horizon) with the strong PatchTST baseline. For the table with full results please see Appendix F.3.

## 5. Limitations

While the Linearizer introduces a principled framework for exact linearization of neural maps, several limitations remain. First, invertible networks are inherently more challenging to train than standard architectures, often requiring careful design. Second, this work represents a broad and general first step: our applications demonstrate feasibility, but none are yet scaled to state-of-the-art benchmarks. Finally, the precise expressivity of the Linearizer remains an open theoretical question.

## 6. Related Work

**Linearization in Dynamical Systems.** Koopman theory linearizes nonlinear dynamics via observables, yielding $z_{t+1} = K z_t$ (discrete) or $\dot{z} = A z$ (continuous) (Mezić, 2005). Data-driven realizations such as DMD and EDMD construct finite-dimensional approximations of this infinite-dimensional operator on a chosen dictionary of observables, which typically produces a square matrix because the dictionary is mapped into itself, although variants with different feature sets exist (Schmid, 2010; Lusch et al., 2018; Mardt et al., 2018). Linear algebraic analyses such as SVD, eigendecomposition, and taking powers are standard in these Koopman approximations. Our work addresses a different object: an arbitrary learned mapping $f : \mathcal{X} \rightarrow \mathcal{Y}$. We learn invertible coordinate maps $g_x$ and $g_y$ and a finite matrix $A$ such that $g_y \circ f \circ g_x^{-1} = A$, which gives the exact finite-dimensional linearity of $f$ between induced spaces. This

formulation allows for distinct input and output coordinates, allows $A$ to be rectangular when $\dim \mathcal{X} \neq \dim \mathcal{Y}$, and transfers linear-algebraic structure in $A$ directly to $f$ in a controlled way. One can compute $\mathrm{SVD}(f)$ via $\mathrm{SVD}(A)$, invert analytically with $A^+$, impose spectral constraints (e.g., idempotency) via $\sigma(A)$, and more.

**Neural Tangent Kernel.** The NTK framework shows that infinitely wide networks trained with small steps evolve linearly in parameter space (Jacot et al., 2018). However, the resulting model remains nonlinear in input-output mapping. In comparison, our contribution achieves input-output linearity in a learned basis for any network width, independent of training dynamics.

**One-Step Diffusion Distillation.** Reducing sampling cost in diffusion models often uses student distillation: Progressive Distillation (Salimans & Ho, 2022), Consistency Models (Song et al., 2023), Distribution Matching Distillation (DMD) and f-distillation fall into this category. A recent work, the Koopman Distillation Model (KDM) (Berman et al., 2025), uses Koopman-based encoding to distill a diffusion model into a one-step generator, achieving strong FID improvements through offline distillation. The Koopman Distillation Model (KDM) distills a pre-trained diffusion teacher into an encoder-linear-decoder for diffusion only. We train from scratch, enforce exact finite-dimensional linearity by construction, and target multiple use cases including composition, inversion, idempotency, and continuous evolution. Unlike some invertible Koopman setups that force a shared map, KDM employs distinct encoder components and a decoder and does not enforce $g_x = g_y$.

**Invertible Neural Networks.** Invertible models like NICE (Dinh et al., 2015), RealNVP (Dinh et al., 2017) and Glow (Kingma & Dhariwal, 2018) use invertible transforms for density modeling and generative sampling. In this work, we make use of such invertible neural networks to impose a bijective change of coordinates that allows linearity (Proposition 2.4 only holds given this invertibility of $g_x$, $g_y$) rather than modeling distribution.

**Group-theoretic and Equivariant Representations.** Let $G$ act on $\mathcal{X}$ via $(g, x) \mapsto g \cdot x$. Many works learn an encoder $e : \mathcal{X} \rightarrow V$ and a linear representation $\rho : G \rightarrow \mathrm{GL}(V)$ such that the equivariance constraint $e(g \cdot x) =$

$\rho(g)\, e(x)$ holds approximately, often with a decoder $d$ that reconstructs $x$ or enforces $d(\rho(g)z) \approx g \cdot d(z)$ (Quessard et al., 2020; Jin et al., 2024). The objective is to make the *action* of a symmetry group linear in latent coordinates. In contrast, we target a different object: an *arbitrary learned map* $f : \mathcal{X} \to \mathcal{Y}$. We construct invertible $g_x, g_y$ and a finite matrix $A$ so that $g_y \circ f \circ g_x^{-1} = A$, which yields *exact* finite-dimensional linearity of $f$ between induced spaces and enables direct linear algebra on $f$ via $A$.

**Manifold Flattening and Diffeomorphic Autoencoders.**
Manifold-flattening methods assume data lie on a manifold $M \subset \mathbb{R}^N$ and learn or construct a map $\Phi : \mathbb{R}^N \to \mathbb{R}^k$ so that $\Phi(M)$ is close to a linear subspace $L$, often with approximate isometry on $M$ (Psenka et al., 2024). Diffeomorphic autoencoders parameterize deformations $\varphi \in \mathrm{Diff}(\Omega)$ with a latent $z$ in a Lie algebra and use a decoder to warp a template, with variants using a log map to linearize the deformation composition (Bône et al., 2019). These approaches *flatten data geometry or deformation fields*, whereas our Linearizer flattens a mapping function rather than a manifold or deformation group.

**Deep Linear Networks.** Networks composed solely of linear layers are linear in the Euclidean basis and are primarily used to study optimization paths (Saxe et al., 2014; Arora et al., 2018; Cohen et al., 2016). They lack expressive power in standard coordinates. In contrast, our Linearizer is expressive thanks to $g_x$ and $g_y$ that are nonlinear over the standard vector spaces, yet maintains exact linearity in the induced coordinate system.

**Normalizing Flows.** Normalizing flows (**??**Rezende & Mohamed, 2015) use invertible networks to map a tractable base distribution to complex data distributions, preserving density via the change-of-variables formula. Linearizers use the same building block, an invertible network, but instead of transporting measures, they transport algebraic structure. In this sense, Linearizers may be viewed as the "linear-algebraic analogue" of normalizing flows: both frameworks exploit invertible transformations to pull back complex objects into domains where they become simple and tractable.

## 7. Conclusion

We introduced the Linearizer, a framework that learns invertible coordinate transformations such that neural networks become exact linear operators in the induced space. This yields new applications in one-step flow matching, modular style transfer, and architectural enforcement of idempotency. Looking ahead, several directions stand out. First, scaling one-step flow matching and IGN to larger datasets and higher resolutions could provide competitive generative models with unprecedented efficiency. Second, the matrix structure of the Linearizer opens the door to modeling motion dynamics: by exploiting matrix exponentials, one could simulate continuous-time evolution directly, ex-

tending the approach beyond generation to physical and temporal modeling. More broadly, characterizing the theoretical limits of Linearizer expressivity, and integrating it with other operator-learning frameworks, remains an exciting avenue for future research. A natural extension is to infinite dimensions: replacing $A$ with a bounded operator on a Hilbert space connects Linearizers to Koopman theory, with the challenge shifting to the expressiveness of $g$. We leave this to future work.

## Acknowledgments

A.S. is a Chaya Fellow, supported by the Chaya Career Advancement Chair. We thank the anonymous ICML 2026 reviewers for their constructive feedback, which substantially improved the presentation of this work. Code is available at `https://github.com/assafshocher/Linearizer`.

## Impact Statement

This paper presents a fundamental architectural framework intended to advance the theoretical understanding and computational efficiency of neural networks. A primary application of our work is accelerating generative diffusion models via one-step sampling. While this offers significant benefits in terms of reduced computational cost and energy consumption during inference, it shares the general societal risks associated with generative AI, such as the potential for creating misleading synthetic content. We believe our specific method does not introduce new ethical concerns beyond those already established in the field of generative modeling.

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

## A. Proof of Proposition 2.3 (Valid Vector Spaces)

*Proof.* We verify that $(V, \oplus_g, \odot_g)$ is a vector space over $\mathbb{R}$ by transporting each axiom via $g$ and $g^{-1}$, using only the definitions

$$u \oplus_g v := g^{-1}(g(u) + g(v)), \qquad a \odot_g u := g^{-1}(a\,g(u)).$$

For $u, v, w \in V$ and $a, b \in \mathbb{R}$:

1. **Closure**

$$u \oplus_g v \;=\; g^{-1}\big(g(u) + g(v)\big) \;\in\; V. \tag{22}$$

2. **Associativity**

$$\begin{aligned}
(u \oplus_g v) \oplus_g w &= g^{-1}\big(g(u \oplus_g v) + g(w)\big) \\
&= g^{-1}\big((g(u) + g(v)) + g(w)\big) \\
&= g^{-1}\big(g(u) + (g(v) + g(w))\big) \\
&= g^{-1}\big(g(u) + g(v \oplus_g w)\big) \\
&= u \oplus_g (v \oplus_g w).
\end{aligned} \tag{23}$$

3. **Commutativity**

$$\begin{aligned}
u \oplus_g v &= g^{-1}\big(g(u) + g(v)\big) \\
&= g^{-1}\big(g(v) + g(u)\big) \\
&= v \oplus_g u.
\end{aligned} \tag{24}$$

4. **Additive identity** (with $0_V := g^{-1}(0)$)

$$u \oplus_g 0_V = g^{-1}\big(g(u) + g(0_V)\big) = g^{-1}\big(g(u) + 0\big) = u. \tag{25}$$

5. **Additive inverse** (with $(-u) := g^{-1}(-g(u))$)

$$u \oplus_g (-u) = g^{-1}\big(g(u) + g(-u)\big) = g^{-1}\big(g(u) - g(u)\big) = g^{-1}(0) = 0_V. \tag{26}$$

6. **Compatibility of scalar multiplication**

$$\begin{aligned}
a \odot_g (b \odot_g u) &= g^{-1}\big(a\,g(b \odot_g u)\big) = g^{-1}\big(a\,(b\,g(u))\big) \\
&= g^{-1}\big((ab)\,g(u)\big) = (ab) \odot_g u.
\end{aligned} \tag{27}$$

7. **Scalar identity**

$$1 \odot_g u = g^{-1}\big(1 \cdot g(u)\big) = g^{-1}\big(g(u)\big) = u. \tag{28}$$

8. **Distributivity over vector addition**

$$\begin{aligned}
a \odot_g (u \oplus_g v) &= g^{-1}\big(a\,g(u \oplus_g v)\big) = g^{-1}\big(a\,(g(u) + g(v))\big) \\
&= g^{-1}\big(a\,g(u) + a\,g(v)\big) = g^{-1}\big(a\,g(u)\big) \oplus_g g^{-1}\big(a\,g(v)\big) \\
&= (a \odot_g u) \oplus_g (a \odot_g v).
\end{aligned} \tag{29}$$

9. **Distributivity over scalar addition**

$$\begin{aligned}
(a + b) \odot_g u &= g^{-1}\big((a + b)\,g(u)\big) = g^{-1}\big(a\,g(u) + b\,g(u)\big) \\
&= g^{-1}\big(a\,g(u)\big) \oplus_g g^{-1}\big(b\,g(u)\big) = (a \odot_g u) \oplus_g (b \odot_g u).
\end{aligned} \tag{30}$$

$\square$

## B. Proof of Proposition 2.9 (SVD of a Linearizer)

Let $f : \mathcal{X} \to \mathcal{Y}$ be a Linearizer

$$f(x) = g_y^{-1}\big(A\, g_x(x)\big), \tag{31}$$

where $g_x : \mathcal{X} \to \mathbb{R}^N$ and $g_y : \mathcal{Y} \to \mathbb{R}^M$ are invertible and $A \in \mathbb{R}^{M \times N}$. Equip $\mathcal{X}$ and $\mathcal{Y}$ with the induced inner products

$$\langle u, v \rangle_{g_x} := \langle g_x(u), g_x(v) \rangle_{\mathbb{R}^N}, \qquad \langle p, q \rangle_{g_y} := \langle g_y(p), g_y(q) \rangle_{\mathbb{R}^M}. \tag{32}$$

Let the (Euclidean) SVD of $A$ be $A = U\Sigma V^\top$, where $U = [u_1, \ldots, u_M] \in \mathbb{R}^{M \times M}$, $V = [v_1, \ldots, v_N] \in \mathbb{R}^{N \times N}$ are orthogonal and $\Sigma = \mathrm{diag}(\sigma_1, \ldots, \sigma_r, 0, \ldots)$ with $\sigma_1 \geq \cdots \geq \sigma_r > 0$. Define

$$\tilde{u}_i := g_y^{-1}(u_i) \in \mathcal{Y}, \qquad \tilde{v}_i := g_x^{-1}(v_i) \in \mathcal{X}. \tag{33}$$

Then $\{\tilde{v}_i\}$ and $\{\tilde{u}_i\}$ are orthonormal sets in $(\mathcal{X}, \langle \cdot, \cdot \rangle_{g_x})$ and $(\mathcal{Y}, \langle \cdot, \cdot \rangle_{g_y})$, respectively, and for each $i \leq r$,

$$f(\tilde{v}_i) = \sigma_i \odot_{g_y} \tilde{u}_i, \qquad \text{and} \qquad f^*(\tilde{u}_i) = \sigma_i \odot_{g_x} \tilde{v}_i, \tag{34}$$

where $f^*$ is the adjoint with respect to the induced inner products and $a \odot_g$ denotes the induced scalar multiplication. Hence $(\{\tilde{u}_i\}_{i=1}^r, \{\sigma_i\}_{i=1}^r, \{\tilde{v}_i\}_{i=1}^r)$ is an SVD of $f$ between the induced Hilbert spaces.

*Proof.* **Orthonormality.** For $i, j$,

$$\langle \tilde{v}_i, \tilde{v}_j \rangle_{g_x} = \langle g_x(\tilde{v}_i), g_x(\tilde{v}_j) \rangle_{\mathbb{R}^N} = \langle v_i, v_j \rangle_{\mathbb{R}^N} = \delta_{ij}, \tag{35}$$

and similarly $\langle \tilde{u}_i, \tilde{u}_j \rangle_{g_y} = \delta_{ij}$. Thus $g_x$ and $g_y$ are isometric isomorphisms from the induced spaces to Euclidean space.

**Action on right singular vectors.** Using $Av_i = \sigma_i u_i$,

$$f(\tilde{v}_i) = g_y^{-1}\big(A\, g_x(\tilde{v}_i)\big) = g_y^{-1}\big(Av_i\big) = g_y^{-1}\big(\sigma_i u_i\big) = \sigma_i \odot_{g_y} \tilde{u}_i. \tag{36}$$

**Adjoint and action on left singular vectors.** The adjoint $f^* : \mathcal{Y} \to \mathcal{X}$ with respect to the induced inner products is characterized by

$$\langle f(x), y \rangle_{g_y} = \langle x, f^*(y) \rangle_{g_x} \quad \text{for all } x \in \mathcal{X}, \ y \in \mathcal{Y}. \tag{37}$$

Transporting through $g_x, g_y$ and using Euclidean adjoints shows that

$$f^*(y) = g_x^{-1}\big(A^\top g_y(y)\big). \tag{38}$$

Therefore, since $A^\top u_i = \sigma_i v_i$,

$$f^*(\tilde{u}_i) = g_x^{-1}\big(A^\top g_y(\tilde{u}_i)\big) = g_x^{-1}\big(A^\top u_i\big) = g_x^{-1}\big(\sigma_i v_i\big) = \sigma_i \odot_{g_x} \tilde{v}_i. \tag{39}$$

**Conclusion.** The triples $\big(\tilde{u}_i, \sigma_i, \tilde{v}_i\big)$ satisfy the defining relations of singular triplets for $f$ between the induced inner product spaces, with $\{\tilde{v}_i\}$ and $\{\tilde{u}_i\}$ orthonormal bases of the right and left singular subspaces. For zero singular values, the same construction holds with images mapped to the null spaces as usual. $\square$

## C. Proof of Proposition 2.7 (Hilbert Space)

Let $g : V \to \mathbb{R}^n$ be a bijection and endow $V$ with the induced vector space operations $(\oplus_g, \odot_g)$ and inner product

$$\langle u, v \rangle_g := \langle g(u), g(v) \rangle_{\mathbb{R}^n}. \tag{40}$$

Then $(V, \langle \cdot, \cdot \rangle_g)$ is a Hilbert space.

*Proof.* By Proposition 2.3, $(V, \oplus_g, \odot_g)$ is a vector space and $g$ is a vector-space isomorphism from $(V, \oplus_g, \odot_g)$ onto $(\mathbb{R}^n, +, \cdot)$. The form $\langle \cdot, \cdot \rangle_g$ is an inner product because it is the pullback of the Euclidean inner product: bilinearity, symmetry, and positive definiteness follow immediately from injectivity of $g$ and the corresponding properties in $\mathbb{R}^n$.

Let $\| \cdot \|_g$ be the norm induced by $\langle \cdot, \cdot \rangle_g$. For any $u, v \in V$,

$$\|u - v\|_g = \|g(u) - g(v)\|_2, \tag{41}$$

so $g$ is an isometry from $(V, \| \cdot \|_g)$ onto $(\mathbb{R}^n, \| \cdot \|_2)$. Hence a sequence $\{u_k\} \subset V$ is Cauchy in $\| \cdot \|_g$ iff $\{g(u_k)\}$ is Cauchy in $\mathbb{R}^n$. Since $\mathbb{R}^n$ is complete, $\{g(u_k)\}$ converges to some $y \in \mathbb{R}^n$. By surjectivity of $g$, there exists $u^\star \in V$ with $g(u^\star) = y$, and then $\|u_k - u^\star\|_g = \|g(u_k) - y\|_2 \to 0$. Thus $(V, \| \cdot \|_g)$ is complete. Therefore $(V, \langle \cdot, \cdot \rangle_g)$ is a Hilbert space. $\square$

## D. Proof of Lemma 2.10 (Pseudo-Inverse of a Linearizer)

**Pseudoinverse of a Linearizer.** Let $f : \mathcal{X} \to \mathcal{Y}$ be a Linearizer $f(x) = g_y^{-1}(A\, g_x(x))$ under the induced inner products $\langle u, v \rangle_{g_x} := \langle g_x(u), g_x(v) \rangle$ on $\mathcal{X}$ and $\langle u, v \rangle_{g_y} := \langle g_y(u), g_y(v) \rangle$ on $\mathcal{Y}$. Write $f^*(y) = g_x^{-1}(A^\mathsf{T} g_y(y))$ for the adjoint and let $A^\dagger$ be the (Euclidean) Moore–Penrose pseudoinverse of $A$ (Penrose, 1955).

The Moore–Penrose pseudoinverse of $f$ with respect to the induced inner products is

$$f^\dagger \;=\; g_x^{-1} \circ A^\dagger \circ g_y. \tag{42}$$

*Proof.* Following Penrose (1955), the Moore–Penrose pseudoinverse of a linear operator is uniquely characterized as the map satisfying four algebraic conditions (the Penrose equations). To establish that $f^\dagger = g_x^{-1} \circ A^\dagger \circ g_y$ is indeed the pseudoinverse of $f$, it therefore suffices to verify these four identities explicitly:

1. $ff^\dagger f = f$,

2. $f^\dagger f f^\dagger = f^\dagger$,

3. $(ff^\dagger)^* = ff^\dagger$,

4. $(f^\dagger f)^* = f^\dagger f$.

We verify the Penrose equations in the induced spaces.

**(1)** $f\, f^\dagger\, f = f$:

$$
\begin{aligned}
f\, f^\dagger\, f &= \left(g_y^{-1} \circ A \circ g_x\right)\left(g_x^{-1} \circ A^\dagger \circ g_y\right)\left(g_y^{-1} \circ A \circ g_x\right)\\
&= g_y^{-1}(AA^\dagger A\, g_x)\\
&= g_y^{-1}(A\, g_x) \;=\; f,
\end{aligned}
\tag{43}
$$

using $AA^\dagger A = A$.

**(2)** $f^\dagger f f^\dagger = f^\dagger$:

$$
\begin{aligned}
f^\dagger f f^\dagger &= \left(g_x^{-1} \circ A^\dagger \circ g_y\right)\left(g_y^{-1} \circ A \circ g_x\right)\left(g_x^{-1} \circ A^\dagger \circ g_y\right)\\
&= g_x^{-1}(A^\dagger A A^\dagger\, g_y)\\
&= g_x^{-1}(A^\dagger g_y) \;=\; f^\dagger,
\end{aligned}
\tag{44}
$$

using $A^\dagger A A^\dagger = A^\dagger$.

**(3)** $(ff^\dagger)^* = ff^\dagger$ on $\mathcal{Y}$:

$$
\begin{aligned}
ff^\dagger &= g_y^{-1}(AA^\dagger\, g_y),\\
(ff^\dagger)^* &= g_y^{-1}((AA^\dagger)^\mathsf{T} g_y) = g_y^{-1}(AA^\dagger\, g_y) = ff^\dagger,
\end{aligned}
\tag{45}
$$

since $AA^\dagger$ is symmetric.

**(4)** $(f^\dagger f)^* = f^\dagger f$ on $\mathcal{X}$:

$$
\begin{aligned}
f^\dagger f &= g_x^{-1}(A^\dagger A\, g_x),\\
(f^\dagger f)^* &= g_x^{-1}((A^\dagger A)^\mathsf{T} g_x) = g_x^{-1}(A^\dagger A\, g_x) = f^\dagger f,
\end{aligned}
\tag{46}
$$

since $A^\dagger A$ is symmetric. All four conditions hold, hence $f^\dagger$ is the Moore–Penrose pseudoinverse of $f$ in the induced inner-product spaces. $\square$

## E. Spectral Invariance under Linear Reparametrization

In Section 3.2 we claimed that, in two regimes used by our applications, the spectral structure of the core $A$ is invariant across reparametrizations of $g$. We make the statement precise here. Throughout, we restrict attention to *linear* reparametrizations of the latent space, i.e. replacements $g \mapsto L \circ g$ for some invertible linear map $L$ on $\mathbb{R}^N$. This is the family of reparametrizations considered in Section 3.2 (e.g. absorbing $Q$ when $A = Q\Lambda Q^{-1}$, or absorbing the SVD orthogonal factors $U, V$). General

nonlinear diffeomorphisms can still keep the core linear in special cases (e.g. coordinate-wise rescalings of a diagonal $A$), but they need not preserve eigenvalues; identifiability under arbitrary diffeomorphisms is therefore strictly weaker than rank-only and is not what our applications rely on.

**Setup.** Suppose two Linearizers with the *same* latent dimension realize the same data-space function $f$, related by a linear reparametrization. Concretely, there exists an invertible linear map $L \in \mathrm{GL}(\mathbb{R}^N)$ with $g_b = L \circ g_a$, and

$$f(x) \;=\; g_a^{-1}\big(A_a\, g_a(x)\big) \;=\; g_b^{-1}\big(A_b\, g_b(x)\big). \tag{47}$$

Substituting $g_b = L \circ g_a$ and matching gives the standard similarity relation

$$A_b \;=\; L\, A_a\, L^{-1}. \tag{48}$$

### E.1. Eigendecomposition invariance when $g_x = g_y$

**Proposition E.1** (Eigendecomposition invariance under $g_x = g_y$)**.** Assume $g_x = g_y = g$ and that the two realizations in Eq. (47) are related by a linear reparametrization $g_b = L \circ g_a$, with $A_a$ diagonalizable. Then $A_a$ and $A_b$ have the same eigenvalues with the same multiplicities, and the data-space eigenvectors of $f$ are uniquely determined: every eigenvector $v$ of $A_a$ corresponds, under both realizations, to the same point in $\mathcal{X}$.

*Proof.* By Eq. (48), $A_b = LA_aL^{-1}$, so $A_a$ and $A_b$ are similar matrices and share their characteristic polynomial; in particular, they share eigenvalues with multiplicities.

Let $A_a v = \lambda v$. Then

$$A_b(Lv) \;=\; (LA_aL^{-1})(Lv) \;=\; LA_a v \;=\; \lambda\, Lv, \tag{49}$$

so $Lv$ is an eigenvector of $A_b$ with the same eigenvalue $\lambda$. The corresponding data-space eigenvectors are

$$g_a^{-1}(v) \quad \text{and} \quad g_b^{-1}(Lv) \;=\; g_a^{-1}(L^{-1}Lv) \;=\; g_a^{-1}(v), \tag{50}$$

which coincide. Hence both eigenvalues and data-space eigenvectors are intrinsic properties of $f$, independent of which realization is used. $\qquad\square$

### E.2. SVD invariance when $g$ is near-isometric

**Proposition E.2** (SVD invariance under near-isometric $g$)**.** Assume the two realizations in Eq. (47) are related by a linear reparametrization $g_b = L \circ g_a$, and that $L$ is orthogonal – i.e. both $g_a$ and $g_b$ act as isometries between $\mathcal{X}$ (with the induced inner product) and $\mathbb{R}^N$ (with the Euclidean inner product). Then the singular values *and* the data-space singular vectors of $f$ are uniquely determined by $f$ alone.

*Proof.* Orthogonality of $L$ means $L^{-1} = L^\top$, so Eq. (48) becomes

$$A_b \;=\; L\, A_a\, L^\top. \tag{51}$$

Let $A_a = U_a \Sigma V_a^\top$ be a singular value decomposition. Then

$$A_b \;=\; L\, U_a\, \Sigma\, V_a^\top\, L^\top \;=\; (LU_a)\, \Sigma\, (LV_a)^\top, \tag{52}$$

and $LU_a, LV_a$ are orthogonal as products of orthogonal matrices. By uniqueness of the SVD (up to permutation/sign on equal singular values), $\Sigma_b = \Sigma$, $U_b = LU_a$, and $V_b = LV_a$. In data space, the $i$-th left singular vector of $f$ under realization $b$ is

$$\tilde{u}_i^{(b)} \;=\; g_b^{-1}(u_b^{(i)}) \;=\; g_b^{-1}(Lu_a^{(i)}) \;=\; g_a^{-1}(L^{-1}Lu_a^{(i)}) \;=\; g_a^{-1}(u_a^{(i)}) \;=\; \tilde{u}_i^{(a)}, \tag{53}$$

and an identical argument applies to the right singular vectors. Hence the full SVD of $f$ is invariant across orthogonal reparametrizations of $g$. $\qquad\square$

**Remark.** The Idempotency application (Section 4.3) explicitly encourages near-isometry of $g$ via $\mathcal{L}_{\mathrm{isometry}}$. In that regime Proposition E.2 guarantees that the binary projector $A$ – including its specific singular vectors, not just its rank – is an intrinsic property of the learned function (up to the small slack tolerated by the regularizer).

# F. Additional Results

## F.1. Style Transfer

We expand the results of style transfer made in the main paper and show multiple more transformations in Figure 6 following the same setup.

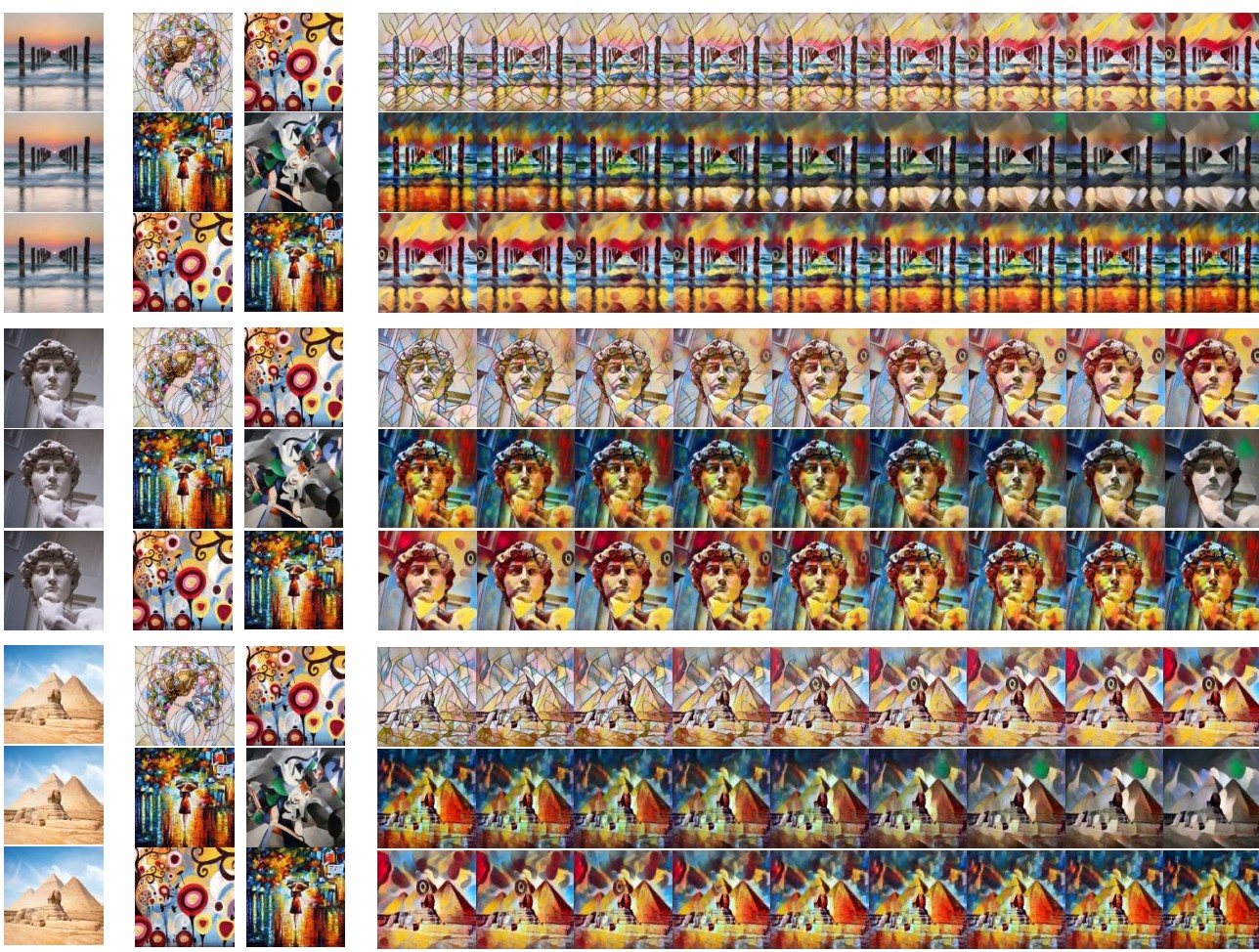

*Figure 6.* **Style transfer examples.** Left: original image. Middle: style transfer using the left-side and right-side style images. Right: interpolation between the two styles.

## F.2. One-step Inverse Interpolation

We extend the results presented in the main text by providing additional interpolation examples on the MNIST (Figure 8) and CelebA (Figure 7) datasets.

## F.3. Additional Applications – Full Results

**CelebFaces Attributes (CelebA).** The dataset includes large pose variation, background clutter, diverse identities, and 40 binary attributes (e.g., *Male*, *Eyeglasses*, *Blond Hair*). After training, our Linearizer attains **87.6%** test accuracy; a ResNet-50 baseline reaches **86.1%**.

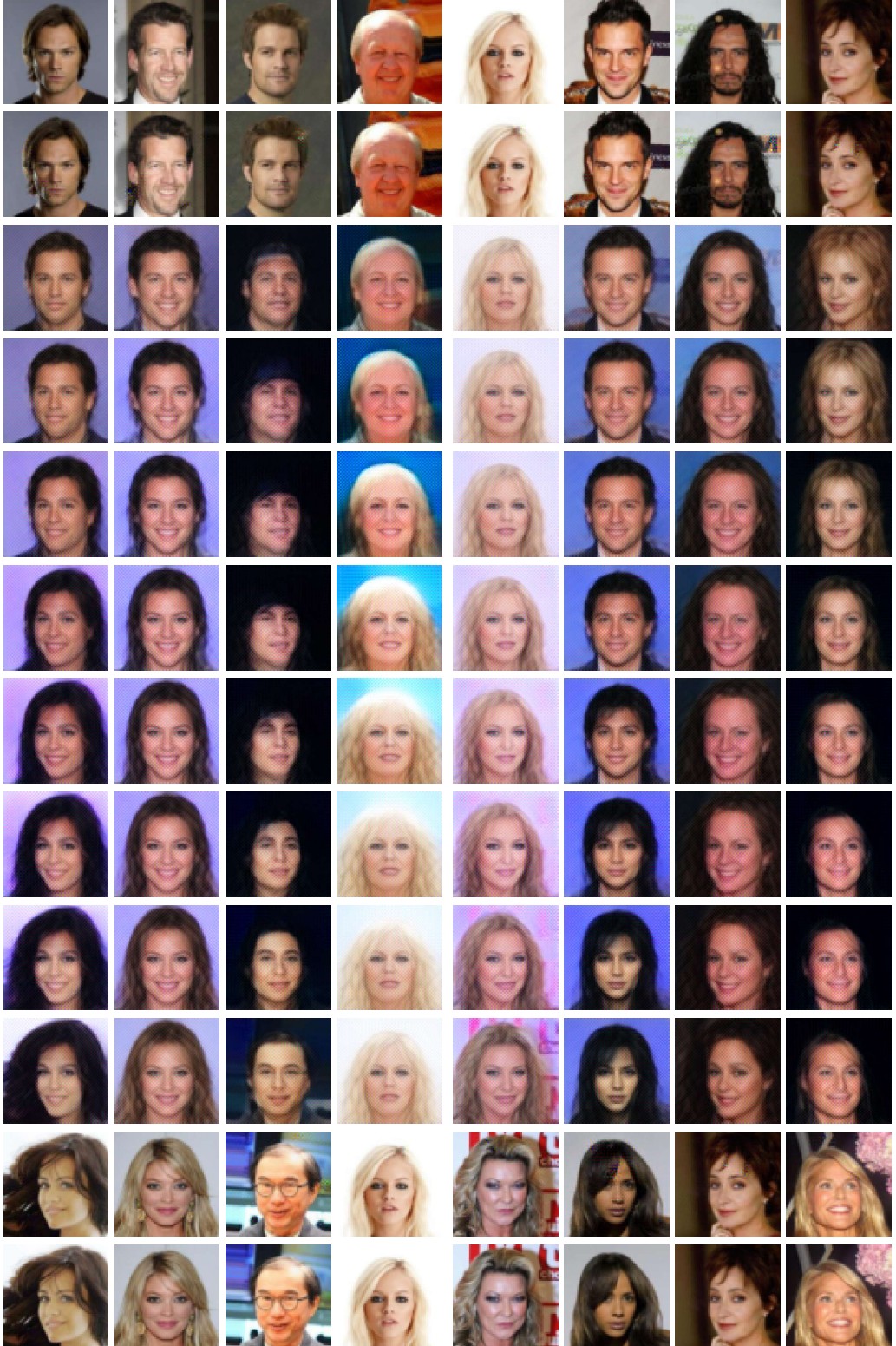

*Figure 7.* Additional interpolations of CelebA via inversion.

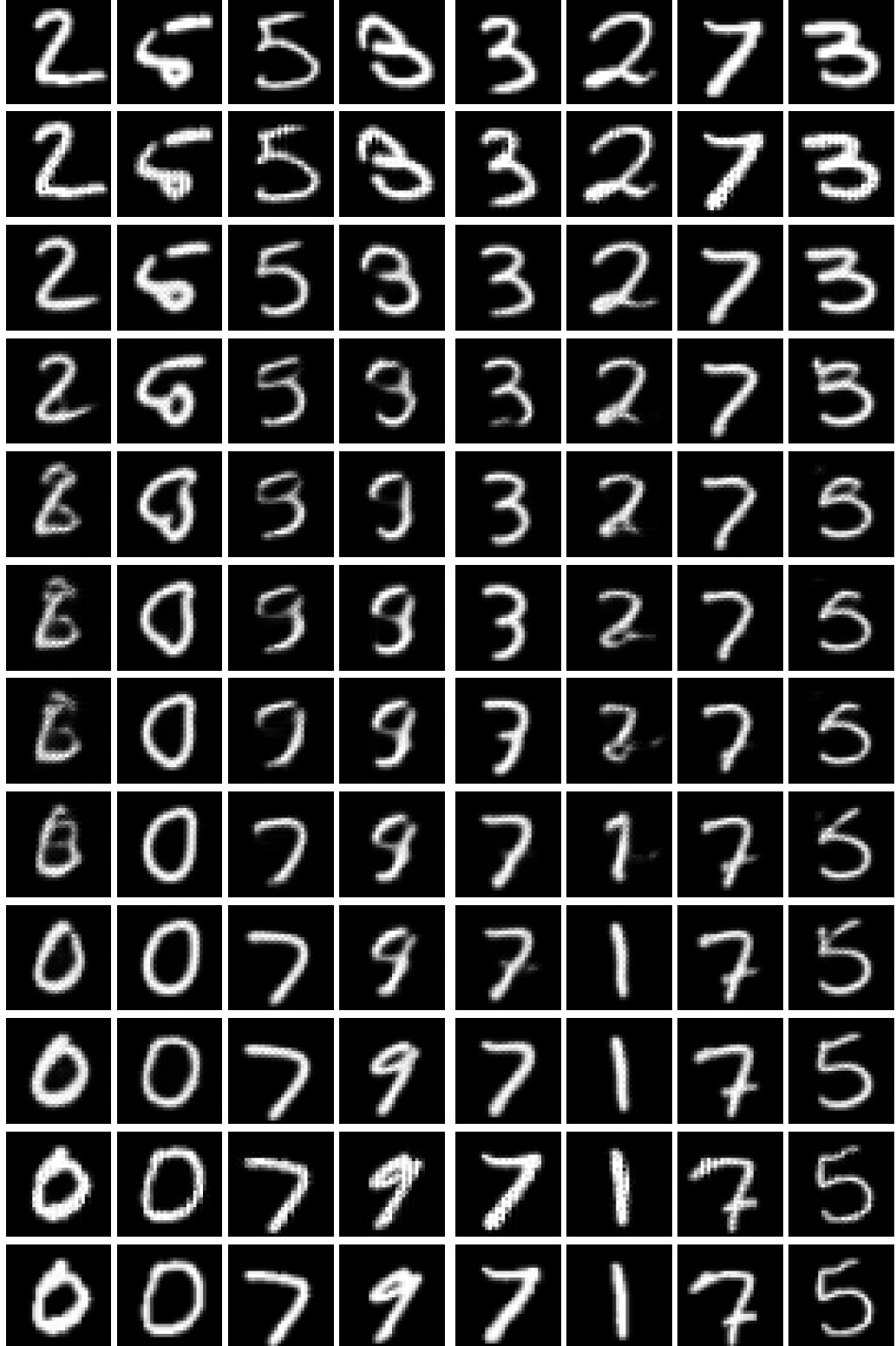

*Figure 8.* Additional interpolations of MNIST via inversion.

**Distillation of nonlinear disentanglement models.** We distill pretrained encoders (VAE, FactorVAE, $\beta$-VAE; checkpoints from the public repository[2]) on DSPRITES and evaluate with **FactorVAE**, **SAP**, and **DCI** (disentanglement, completeness, informativeness). The Linearizer closely matches the original encoders (rows prefixed with *Linearizer-*).

*Table 2.* Disentanglement metrics on DSPRITES. Higher is better ($\uparrow$).

| Model | FactorVAE $\uparrow$ | SAP $\uparrow$ | DCI-D $\uparrow$ | DCI-C $\uparrow$ | DCI-I $\uparrow$ |
|---|---|---|---|---|---|
| VAE | .28±.03 | .02±.01 | .49±.02 | .27±.01 | .94±.00 |
| Linearizer-VAE | .29±.03 | .02±.00 | .49±.01 | .27±.01 | .93±.01 |
| FactorVAE | .33±.01 | .06±.01 | .64±.01 | .51±.01 | .94±.00 |
| Linearizer-FactorVAE | .35±.03 | .05±.01 | .64±.01 | .50±.00 | .94±.00 |
| BetaH | .59±.05 | .23±.01 | .83±.02 | .71±.00 | .95±.00 |
| Linearizer-BetaH | .69±.05 | .24±.01 | .80±.00 | .71±.00 | .95±.00 |

**Weather Prediction.** We evaluate multi-step regression where the model predicts future values given 96 past steps. We compare against **Autoformer** (Wu et al., 2021) and **PatchTST** (Nie, 2022). Columns denote horizons (steps ahead); scores are MSE (lower is better).

*Table 3.* Weather forecasting (MSE; lower is better) at different horizons.

| Model | 96 | 192 | 336 | 720 |
|---|---|---|---|---|
| Linearizer | 0.167 | 0.224 | 0.257 | 0.368 |
| PatchTST | 0.159 | 0.211 | 0.241 | 0.334 |
| Autoformer | 0.247 | 0.321 | 0.341 | 0.411 |

# G. Derivation for Runge–Kutta Collapse to One Step

We follow the setup in Sec. 4.1. Let $z_t := g(x_t)$ and $f(x,t) = g^{-1}(A_t\, g(x))$. In the induced coordinates, addition and scaling are Euclidean, so the ODE update acts linearly on $z_t$.

**RK4 step in the induced space.** Define $t_n = n\Delta t$ and $z_n := z_{t_n} = g(x_{t_n})$. The classical RK4 step from $t_n$ to $t_{n+1} = t_n + \Delta t$ is

$$k_1 = A_{t_n}\, z_n, \tag{54}$$

$$k_2 = A_{t_n + \frac{\Delta t}{2}}\left(z_n + \tfrac{\Delta t}{2}\, k_1\right), \tag{55}$$

$$k_3 = A_{t_n + \frac{\Delta t}{2}}\left(z_n + \tfrac{\Delta t}{2}\, k_2\right), \tag{56}$$

$$k_4 = A_{t_n + \Delta t}(z_n + \Delta t\, k_3), \tag{57}$$

$$z_{n+1} = z_n + \tfrac{\Delta t}{6}\left(k_1 + 2k_2 + 2k_3 + k_4\right). \tag{58}$$

Since each $k_i$ is linear in $z_n$, there exists a matrix $M_n$ such that $z_{n+1} = M_n z_n$. Expanding the linearity gives the explicit one-step operator

$$\begin{aligned}
M_n = I + \tfrac{\Delta t}{6}\Big[ & A_{t_n} + 2\, A_{t_n + \frac{\Delta t}{2}}\Big(I + \tfrac{\Delta t}{2}\, A_{t_n}\Big) \\
& + 2\, A_{t_n + \frac{\Delta t}{2}}\Big(I + \tfrac{\Delta t}{2}\, A_{t_n + \frac{\Delta t}{2}}\big(I + \tfrac{\Delta t}{2}\, A_{t_n}\big)\Big) \\
& + A_{t_n + \Delta t}\Big(I + \Delta t\, A_{t_n + \frac{\Delta t}{2}}\big(I + \tfrac{\Delta t}{2}\, A_{t_n + \frac{\Delta t}{2}}\big(I + \tfrac{\Delta t}{2}\, A_{t_n}\big)\big)\Big)\Big].
\end{aligned} \tag{59}$$

**Collapsed one-step operator.** Iterating $N$ steps and collecting the product in latent space yields

$$B := \prod_{n=0}^{N-1} M_n, \qquad \text{so that} \qquad g(\hat{x}_1) = B\, g(x_0), \quad \hat{x}_1 = g^{-1}\big(B\, g(x_0)\big). \tag{60}$$

---

[2] https://github.com/YannDubs/disentangling-vae

The last two equations ((59) and (60)) are the RK4 analogues of the Euler collapse in the main text: the multi-step solver is replaced by a single multiplication with $B$ computed once after training.

**Time-invariant sanity check (optional).** If $A_t \equiv A$ is constant, then (59) reduces to the degree-4 Taylor polynomial of $e^{\Delta t A}$:

$$M_n \;=\; I + \Delta t\, A + \frac{(\Delta t)^2}{2}\, A^2 + \frac{(\Delta t)^3}{6}\, A^3 + \frac{(\Delta t)^4}{24}\, A^4, \qquad B \;=\; M^N. \tag{61}$$

This matches the behavior of classical RK4 on linear time-invariant systems.

## H. Implementation Details

### H.1. Backbones and Block Structure

We use **6 invertible blocks** in all models. The two architectures differ only in the internal block type:

| Model family | Per-block flow (forward) |
|---|---|
| Diffusion / Style | (optional Squeeze2x2 for 1ch) $\rightarrow$ ActNorm $\rightarrow$ Affine Coupling $(x_1 \,|\, x_2) \rightarrow$ Affine Coupling $(x_2 \,|\, y_1) \rightarrow$ Invertible $1{\times}1$ Conv |
| IGN | SpatialSplit2x2Rand $\rightarrow$ Additive Couplings with $\{F, G\} \rightarrow$ concat $\rightarrow$ PixelShuffle $(2\times)$ |

*Table 4.* Block-level flow for the two architectures; repeated 6 times.

**Notes.**

- **ActNorm**: per-channel affine with data-dependent init on first batch.

- **Affine coupling (Diffusion/Style)**: shift/log-scale predicted by a tiny U-Net conditioner; clamped $\log s$.

- **Invertible $1{\times}1$ conv (Diffusion/Style)**: learned channel permutation/mixer (orthogonal init).

- **SpatialSplit2x2Rand (IGN)**: deterministic $2{\times}2$ space partition into two streams; inverse merges.

- **PixelShuffle / unshuffle (IGN)**: reversible $2\times$ spatial reindexing between blocks.

- **We do not use `InvTanhScaled`**.

**The blocks:**

1. **Diffusion/Style.** We use a U-Net (Ronneberger et al., 2015), adapting the implementation of Karras et al. (2022). We modify only two hyperparameters: `model_channels = 16` and `channel_mult = [1, 1]`.

2. **IGN.** We use bottleneck architecture, gradually downscaling to spatial size $1 \times 1$ by stride 2 convolutions, and then using transposed convolutions back to original resolution. See table below for exact layers.

### H.2. Time Path (Diffusion)

In diffusion/flow-matching models, the scalar $t$ is embedded and fed to: (i) the affine-coupling conditioners inside $g$, and (ii) the time-dependent core $A_t$ (low-rank MLP). Style and IGN models do not use $t$.

### H.3. Cores $A$

- **Diffusion**: Two non-linear MLPs that produce two matrices for the low-rank parameterization $A = A_1 A_2$ with $\mathrm{rank}(A) = 16$.

- **Style.** A kernel $4 \times 4$ is produced by a non-linear hypernetwork $L(\text{style}) = A_{\text{ker}}^{\text{style}}$, which we then apply to the image via a 2D convolution (PyTorch).

- **IGN**: diagonal $A$ (elementwise scale in latent); for idempotent IGN we threshold the diagonal with STE.

## H.4. Conditioners

**Diffusion / Style (Affine-coupling conditioners).** Each coupling uses a tiny U-Net that maps $C_{\text{in}} = C/2 \to 2\,C_{\text{in}}$ (shift, $\log s$).

| # | Layer (per conditioner) |
|---|---|
| 1 | GroupNorm $\to$ Conv3 $\times$ 3 $\to$ SiLU |
| 2 | Residual block $\times$2 (GroupNorm, Conv3 $\times$ 3, SiLU) |
| 3 | Final Conv3 $\times$ 3 to $2\,C_{\text{in}}$; clamp $\log s$ |

*Table 5.* Minimal U-Net-style conditioner used in affine couplings.

**IGN (Additive-coupling CNNs).** Each coupling uses a plain CNN (*no U-Net skip concatenations*); forward is additive, inverse is exact.

| # | **F** or **G** layer stack |
|---|---|
| 1 | Conv2d $2 \to 8$, kernel 4, stride 2 |
| 2 | Conv2d $8 \to 32$, kernel 4, stride 2 |
| 3 | Conv2d $32 \to 128$, kernel 4, stride 2 |
| 4 | Conv2d $128 \to 512$, kernel 4, stride 2 |
| 5 | ConvTranspose2d $512 \to 128$, kernel 4, stride 2 |
| 6 | ConvTranspose2d $128 \to 32$, kernel 4, stride 2 |
| 7 | ConvTranspose2d $32 \to 8$, kernel 4, stride 2 |
| 8 | ConvTranspose2d $8 \to 2$, kernel 4, stride 2 $\to$ $\tanh$ |

*Table 6.* CNN used for $F$ and $G$ inside IGN additive couplings.

## H.5. Losses (weights fixed)

**IGN.**

$$\lambda_{\text{rec}} = 1.0, \quad \lambda_{\text{sparse}} = 0.75, \quad \lambda_{\text{iso}} = 0.001.$$

We use (i) reconstruction MSE, (ii) sparsity/rank on the diagonal of $A$, and (iii) a light isometry term on $g$.

**Diffusion / Flow-Matching.** We use the main training loss described in the paper. For additional stability, we empirically found that adding the perceptual reconstruction terms $d(x_0,\, g^{-1}(g(x_0)))$ and $d(x_1,\, g^{-1}(g(x_1)))$ (with $d$ as LPIPS (Zhang et al., 2018)), together with the alignment term $\|A\,g(x_0) - g(x_1)\|^2$, yields more stable training and higher-quality models. The primary loss in the paper also uses LPIPS as the distance metric. No explicit weighting between losses was required.

## H.6. Training-stability monitoring

Although $g$ is invertible by architecture, finite-precision arithmetic and ill-conditioned couplings can still degrade numerical invertibility during training. To detect this we monitor two quantities throughout optimization:

- **Linearization consistency.** For random $a_1, a_2 \in \mathbb{R}$ and random $x_1, x_2$ from the data batch, we check that

$$f(a_1 \odot_x x_1 \oplus_x a_2 \odot_x x_2) \approx a_1 \odot_y f(x_1) \oplus_y a_2 \odot_y f(x_2),$$

  i.e. that the equality from Proposition 2.4 holds numerically. A growing gap signals drift in $g_x, g_y$ or $A$.

- **Invertibility residual.** We track $\|g(g^{-1}(z)) - z\|$ and $\|g^{-1}(g(x)) - x\|$ on held-out samples; both should remain at machine-precision levels.

For numerical safety, we additionally include an explicit *invertibility loss* $\mathcal{L}_{\text{inv}} = \|g^{-1}(g(x)) - x\|^2 + \|g(g^{-1}(z)) - z\|^2$ (small weight) during diffusion/FM training. This is redundant with the architectural guarantee but stabilizes mixed-precision training and makes deviations easy to flag.

## H.7. One-line Hyperparameter Summary

See Table.7.

| Item | Setting |
|------|---------|
| Invertible blocks | 6 (both families) |
| Permutations | Diff/Style: invertible $1{\times}1$ conv; IGN: SpatialSplit2x2Rand + PixelShuffle |
| ActNorm | Enabled (both) |
| Core $A$ | Diff/Style: low-rank 16; IGN: diagonal (STE for projector) |
| Time $t$ | Used in Diffusion/Style (conditioners + $A_t$); not used in IGN |

*Table 7.* Everything needed to reproduce the scaffolding.

# I. Linearizer Universal Final Fit Theorem

**New vs. existing.** The two ingredients used below are classical and not novel: Fact I.2 (smooth diffeomorphism extension on finite sets) follows from standard results on diffeomorphism groups, and Fact I.3 is a paraphrased statement of the INN universality result of Teshima et al. (2020). The novel contribution of this appendix is the *construction* that combines them: Lemma I.4 (existence of an ideal, Linearizer-realizable interpolant for any finite dataset) and Theorem I.5 (the corresponding $\varepsilon$-approximation guarantee when the diffeomorphisms are realized by INNs). Lemma I.4 is, to our knowledge, new – it is what makes the linear-bottleneck architecture compatible with universal approximation. Theorem I.5 is then a routine continuity-and-compactness argument given these two facts; we include the full proof for completeness.

**Definition I.1** (Linearizer-realizable map). A function $F^* : \mathbb{R}^N \to \mathbb{R}^M$ is *Linearizer-realizable* if there exist a linear map $A^* : \mathbb{R}^N \to \mathbb{R}^M$ and diffeomorphisms $g_x^* : \mathbb{R}^N \to \mathbb{R}^N$ and $g_y^* : \mathbb{R}^M \to \mathbb{R}^M$ such that

$$F^*(x) \;=\; (g_y^*)^{-1}\big(A^* \, g_x^*(x)\big) \qquad \text{for all } x \in \mathbb{R}^N.$$

**Fact I.2** (Diffeomorphism Extension on Finite Sets). Let $X = \{x_1, \ldots, x_n\} \subset \mathbb{R}^N$ be a finite set of $n$ distinct points, and let $P = \{p_1, \ldots, p_n\} \subset \mathbb{R}^N$ be any other finite set of $n$ distinct points. There exists a global diffeomorphism $g^* : \mathbb{R}^N \to \mathbb{R}^N$ such that $g^*(x_i) = p_i$ for all $i = 1, \ldots, n$.

**Fact I.3** (INN Approximation, Teshima et al. 2020; informal). Let $K \subset \mathbb{R}^d$ be compact, and let $h : \mathbb{R}^d \to \mathbb{R}^d$ be a $C^1$ diffeomorphism. For any $\eta > 0$, there exists an invertible neural network (INN) $H_\eta : \mathbb{R}^d \to \mathbb{R}^d$ such that $\sup_{z \in K} \|H_\eta(z) - h(z)\| < \eta$ and $\sup_{z \in h(K)} \|H_\eta^{-1}(z) - h^{-1}(z)\| < \eta$.

**Lemma I.4** (Existence of a Realizable Fit). *Let $X = \{x_1, \ldots, x_n\} \subset \mathbb{R}^N$ be a finite set of distinct data points, and let $Y = \{y_1, \ldots, y_n\} \subset \mathbb{R}^M$ be the corresponding target points. There exists an ideal, Linearizer-realizable function $F^*$ that perfectly fits the data, i.e., $F^*(x_i) = y_i$ for all $i$.*

*Proof.* Let $P = \{p_1, \ldots, p_n\} \subset \mathbb{R}^N$ be a set of $n$ distinct latent points lying on the first coordinate axis, defined as $p_i = (i, 0, \ldots, 0)^\top$.

We construct the linear map $A^* : \mathbb{R}^N \to \mathbb{R}^M$ to preserve the distinctness of these points by mapping the first axis of the domain to the first axis of the codomain. We distinguish two cases based on the dimensions:

- **Case 1** ($M \leq N$)**:** We define $A^*$ as the projection onto the first $M$ coordinates. For any vector $v = (v_1, \ldots, v_N)^\top \in \mathbb{R}^N$:

$$A^*(v) = (v_1, \ldots, v_M)^\top \in \mathbb{R}^M.$$

- **Case 2** ($M > N$)**:** We define $A^*$ as the canonical embedding (zero-padding) into the first $N$ coordinates. For any vector $v = (v_1, \ldots, v_N)^\top \in \mathbb{R}^N$:

$$A^*(v) = (v_1, \ldots, v_N, \underbrace{0, \ldots, 0}_{M-N})^\top \in \mathbb{R}^M.$$

In both cases, we define the target latent points as $Q = \{q_1, \ldots, q_n\}$ where $q_i = A^* p_i$. Since the points $p_i$ differ in their first coordinate ($i$), and $A^*$ preserves the first coordinate in both cases (as $M, N \geq 1$), the resulting points $q_i$ are distinct in $\mathbb{R}^M$.

By Fact I.2, there exists a global diffeomorphism $g_x^* : \mathbb{R}^N \to \mathbb{R}^N$ such that $g_x^*(x_i) = p_i$ for all $i$. Similarly, there exists a global diffeomorphism $g_y^* : \mathbb{R}^M \to \mathbb{R}^M$ such that $g_y^*(y_i) = q_i$ for all $i$.

We define the ideal Linearizer $F^*$ as:
$$F^*(x) := (g_y^*)^{-1}(A^* g_x^*(x)).$$

Checking this function on the data points $x_i$:
$$F^*(x_i) = (g_y^*)^{-1}(A^* p_i) = (g_y^*)^{-1}(q_i) = y_i.$$

Thus, a Linearizer-realizable function $F^*$ exists that perfectly fits the finite dataset. $\qquad\square$

**Theorem I.5** (A Linearizer can fit any finite dataset). *Let $X = \{x_1, \ldots, x_n\} \subset \mathbb{R}^N$, $N \geq 2$, be a finite set and $F : X \to \mathbb{R}^M$ be the target function. For every $\varepsilon > 0$, there exist invertible neural networks $G_x, G_y$ and a linear map $A$ such that the Linearizer $\hat{F}(x) := G_y^{-1}(A\, G_x(x))$ achieves arbitrarily low error on the dataset:*

$$\sup_{x \in X} \|\hat{F}(x) - F(x)\| < \varepsilon.$$

*Proof.* By Lemma I.4, there exists an ideal, Linearizer-realizable function $F^*(x) = (g_y^*)^{-1}(A^* g_x^*(x))$ such that $F(x) = F^*(x)$ for all $x \in X$. We need to prove that our INN-based architecture $\hat{F}$ can approximate $F^*$ on the compact set $X$.

Let $\varepsilon > 0$ be given. Set $A := A^*$. The proof follows the standard $\epsilon$-$\delta$ error decomposition.

$$
\begin{aligned}
\hat{F}(x) - F^*(x) &= G_y^{-1}\big(A^* G_x(x)\big) - (g_y^*)^{-1}\big(A^* g_x^*(x)\big) \\
&= \underbrace{G_y^{-1}\big(A^* G_x(x)\big) - (g_y^*)^{-1}\big(A^* G_x(x)\big)}_{(I)} + \underbrace{(g_y^*)^{-1}\big(A^* G_x(x)\big) - (g_y^*)^{-1}\big(A^* g_x^*(x)\big)}_{(II)}.
\end{aligned}
$$

The ideal diffeomorphisms $g_x^*$ and $(g_y^*)^{-1}$ are continuous and are being approximated on compact sets. By Fact I.3, we can choose INNs $G_x$ and $G_y$ that are arbitrarily close to $g_x^*$ and $g_y^*$ (and their inverses) on these sets.

As both $(g_y^*)^{-1}$ and $A^*$ are continuous (and uniformly continuous on any compact set), we can make the norms of both terms (I) and (II) arbitrarily small by choosing sufficiently accurate INN approximations $G_x$ and $G_y$ (i.e., for a small enough $\delta$). We can thus choose $\delta$ such that the total error is $< \varepsilon$. $\qquad\square$

# J. Additional Illustrations

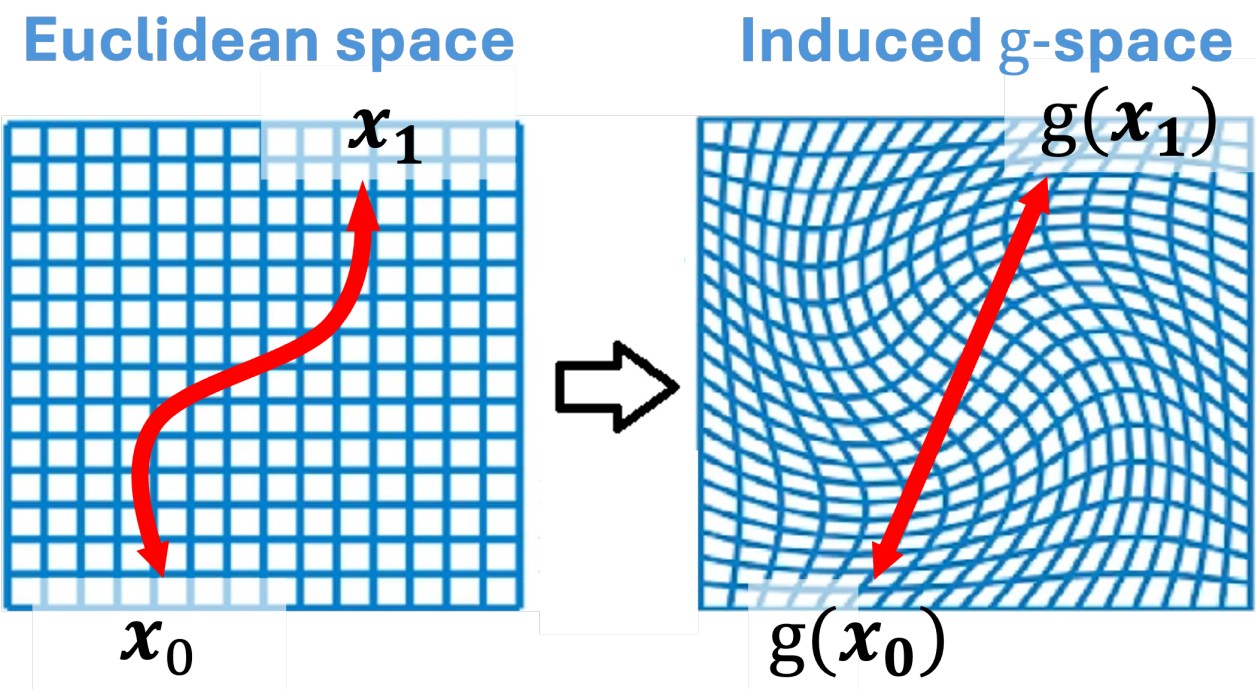

*Figure 9.* **Conceptual illustration of the geometric intuition.** We visualize the Linearizer in the context of Flow Matching (where $g_x = g_y = g$). (Left) The data manifold in standard Euclidean space. The optimal transport path between noise $x_0$ and data $x_1$ is a non-linear curve. (Right) The induced $g$-space. The network learns a coordinate transformation where this complex path is rectified into a simple straight line (a geodesic), enabling exact one-step generation.

