# OpenReview forum: "Who Said Neural Networks Aren't Linear?"
_ICML.cc/2026/Conference — ICML 2026 regular_

### Official Review · Reviewer_92o5 · 2026-02-17

**Soundness:** 3
**Presentation:** 3
**Significance:** 3
**Originality:** 2
**Overall Recommendation:** 3
**Confidence:** 5

**Summary:**

This paper introduces the Linearizer framework where a non-linear mapping $f:\mathbb{R}^N\to \mathbb{R}^M$ is expressed as $f(x)=\phi^{-1}(A\theta (x)) $, where $\phi$ and $\theta$ are learned invertible transformations and $A$ is a linear map. Under this formulation, the function $f$ becomes linear in the induced latent coordinates.

The authors study the expressive power of this representation and show that under suitable assumptions on $\phi$ and $\theta$ (like diffeomorphisms or invertible neural networks) the model can approximate broad classes of functions. The paper also analyzes the role of $A$, showing that in the general setting its role is largely determined by its rank, while additional structure may arise in constrained formulations when $\phi=\theta$.

Finally, the paper discusses several applications (e.g. one-step flow matching, modular styler transfer and linear idempotent generative networks) demonstrating how complex transformations can be realized via linear operations in learned coordinate systems.

**Compliance With Llm Reviewing Policy:**

Affirmed.

**Final Justification:**

I thank the authors for their rebuttal. While I appreciate the clarifications, my main theoretical questions remain largely unresolved, particularly regarding the intrinsic role of the linear operator $A$, the identifiability of its structure beyond rank, and the extent to which the learned linearization reflects more than a flexible reparameterization. The rebuttal provides intuition and examples, but does not address these points in a principled way.

Therefore, I am not able to recommend this paper for acceptance. My overall recommendation remains unchanged: 3 (weak reject), with confidence 5.

**Key Questions For Authors:**

1. the analysis in Section 3.2 suggests that in the general setting with different invertible maps, the effect of $A$ can largely be reduced (up to reparametrization) to its rank. Are tehre settings where properties beyond rank (like spectral structure) have a measurable impact on performance or representation quality?

2. Given that $\phi$ and $\theta$ are highly expressive invertible networks (with potential high computational cost), what is the concrete advantage of introducing the linear operator $A$, compared to directly learning a standard nonlinear model?

3. The paper emphasizes that the model is linear in learned coordinates and to me is not completely clear the meaningfulness of linearization (as shown in the paper). To what extent is the linearization non-trivial, rather than a consequence of flexible reparametrization? Are tehre constrains on $\phi$ and $\theta$ under which the linear representation becomes unique or canonical?

4. Do different parameterizations lead to different learned structures, or are they equivalent I practice?

5. The framework resembles a conjugation of operators. Do the authors envision a meaningful extension to infinite-dimensional settings where the linear operator has non-trivial spectral structure? If so, what additional constrains would be required for the linearization to carry meaningful information?

**Limitations:**

The proposed representation may not be identifiable, as different invertible coordinate changes may yield equivalent functions, making the learned linearization dependent on arbitrary choices. Clarifying this point would strengthen the paper. In the general setting, much of the structure of $A$ can be absorbed into the nonlinear maps, limiting its intrinsic role beyond rank. The approach also requires learning INNs, which increases computational cost. Finally, the empirical evaluation does not clearly demonstrate that the linear formulation provides consistent advantages or isolates the contribution of the linear component.

**Strengths And Weaknesses:**

Why originality is fair? The paper combines known ideas in a coherent way, but does not introduce a clearly new principle beyond reparametrization. For example, the central idea of representing functions as a composition of $\phi^{-1}\circ A \circ \theta $ is essentially a conjugacy formulation which is classical in dynamical systems.

Strengths: The paper provides a clean and unified framework for representing nonlinear mappings via linear operators in learned coordinates. This viewpoint is intuitive and connects to classical ideas as conjugacy in dynamical systems. A theoretical analysis of the linear core is given (even if debatable) which helps clarify what information $A$ can encode. Expressiveness results are given helping positioning the framework relative to universal approximation, building upon invertible neural networks (INNs) so important in generative modeling. Finally, the paper explores several application domains showing the flexibility of the framework and its potential applicability.

Weaknesses: I have several major concerns that I outline here below.

1. The representation proposed in the paper can be interpreted as a change of coordinates formulation that seems to me a reparametrization statement (which is by the way well-known in areas like dynamical systems). So, it is unclear to me to what extent this is a fundamentally new modeling paradigm.

2. Section 3.2 shows that in the general setting $A$ reduces to its rank. this suggests that most of the modeling power resides on the nonlinear maps $\phi$ and $\theta$ so that the linear component $A$ may not capture meaningful intrinsic structure of the function.

3. The framework allows very flexible invertible transformations like diffeomorphisms or INNs. Without additional constraints the linearization may be largely non-unique and dependent on arbitrary coordinate choices, which limits theoretical insights.

4. While the model establishes that the model is expressive, it is less clear whether such representations provide intrinsic structural information about the function. In particular, the formulation does not  identify invariants that are independent of the chosen reparamatrization.

5. The approach requires learning INNs which can be computationally demanding and architecturally constrained. Given that the experimental section does not clearly demonstrate that the proposed formulation provides advantages over existing methods and that much of the complexity appears to reside in the nonlinear maps (as noted in point 2. above), it is not clear that enforcing linearity in latent space yields a practical advantage over standard architectures.

Minor Concerns (Readability and Precision):

-- Some statements lack precision or are potentially misleading. For example, the claim that "linear subspaces are either trivial or infinite-dimensional" Subsection 3.1, page 3, lines 150-151, is incorrect in finite-dimensional settings and should be clarified. I suggest to rewrite the entire Subsection 3.1 Expressiveness, where key arguments could be stated more directly and formally.

-- In Subsection 3.2, page 4, line 197, that sentence seems unfinished. And later in line 202 of the same page there is a typo, it should read $g_x, g_y$.

-- in Appendix H, Fact H.2 would benefit from an explicit reference. Also, the authors should specify which parts of Lemma H.4 and Theorem H.5 are new versus direct consequences of existing results on INNs or diffeomorphisms.


Overall, I find the paper to present an interesting and unifying perspective. However, the current version raises several concerns regarding teh novelty of the theoretical contributions, the intrinsic role of the linear operator and the strength of the empirical validation. I therefore leans towards a weak reject, and encourage the authors to further strengthen both the theoretical analysis and the empirical validation.

---

> ### Author Rebuttal · Authors · 2026-03-28
>
> We thank the reviewer for the careful reading and the important observations.
>
> &nbsp;
>
> # "... a composition of $g^{-1} \circ A \circ g$ is essentially a conjugacy..."
> Our contribution is bringing Transfer of Structure, using conjugacy, to the data-driven setting. Specifically:
> 1. A **method to learn the conjugacy from data**, where neither $g$ nor $A$ are given. Realizing the model using INNs and a low rank matrix, training end-to-end.
> 2. **Applications** enabled by combining conjugacy with learnability: one-step FM (Eq. 15), modular style transfer (Eq. 18), and global idempotency (Lemma 4.2).
> 3. An **expressiveness proof** (Theorems H.4, H.5): fits any finite dataset to arbitrary precision.
> 4. A **complete derived toolkit** in **finite dimensions** for **learned** mappings: SVD, pseudoinverse, transpose, Hilbert space structure.
>
> &nbsp;
>
> # "...may be largely non-unique...", "... not identify invariants that are independent...", "...settings where properties beyond rank (like spectral structure) have a measurable impact..."
> Our demonstrated applications use either $g_x = g_y$ (E.g., FM) or near-isometric $g$ (Eq. 21), and in both cases the invariants go beyond rank, with **full unique spectral decompositions**.
>
> - **$\mathbf{g_x = g_y}$**: the **full eigendecomposition** in data space is invariant. Given two realizations
> $$f = g^{-1} \circ A \circ g$$
> $$f = g'^{-1} \circ A' \circ g'$$
> $$ $$
> $$A' = (g' \circ g^{-1})A(g' \circ g^{-1})^{-1} \implies \text{same eigenvalues}$$
> $$\tilde{v}'_i = g'^{-1}((g' \circ g^{-1})(v_i)) = g^{-1}(v_i) = \tilde{v}_i \implies \text{same eigenvectors (up to the usual scaling freedom in the induced space)}$$
>
> &nbsp;
>
> - **$\mathbf{g}$ is near-isometric ($\mathbf{\|g(x)\| = \|x\|}$)**: the **full SVD** is invariant: $h_x, h_y$ are orthogonal, and the same cancellation applies to singular values and vectors.
>
> In both regimes, the decomposition is as unique as in the standard Euclidean-linear case. We will add this to the paper.
>
> &nbsp;
>
> # "...What is the concrete advantage of introducing the linear operator $A$, compared to ..."
> Structural guarantees no standard architecture provides: exact one-step generation (not distillation), exact pseudo-inversion via $A^\dagger$, exact idempotency by constraining $A$'s eigenvalues, and compositionality (Proposition 2.5). These are mathematical properties of the architecture. It also opens the door to spectral analysis of nonlinear learned mappings.
>
> Importantly, we have extended the experiment in Figure 4a by replacing the linear operator $A$ with a nonlinear MLP of comparable capacity, trained directly for one-step generation. FID increases to 157. This confirms that the linearity of $A$ is essential: the one-step collapse (Eq. 15) relies on closure of linear maps under composition, a property MLPs lack. Despite having greater capacity, the MLP produces worse results.
>
> &nbsp;
>
> # "...most of the modeling power resides on the nonlinear maps..."
> We agree that $g$ carries most of the expressiveness. However, $A$'s role is structural and unreducible (In most applications, $g_x = g_y$, then $A = I \Rightarrow f(x) = g^{-1}(g(x)) = x$). Furthermore, most applications use families of A matrices (in FM, a continuous family parameterized by t). Its role per application:
>
> - **FM:** $A_t$ encodes time-dependent velocity ($g$ has no $t$ input). The full spectral structure of each $A_t$ matters, not just rank. One-step collapse (Eq. 15) follows from linearity of $A$.
> - **Idempotency:** $A$ is a binary projector; its rank determines the projection dimension.
> - **Style Transfer:** different styles = different $A$, shared $g$. Modularity comes entirely from $A$.
>
> &nbsp;
>
> # "Does the framework resemble a conjugation of operators... meaningful extension to infinite-dimensional settings where the linear operator has non-trivial spectral structure?"
> Yes. In the constrained regime, eigenvalues of $A$ carry spectral information (Section 3.2). An infinite-dimensional extension would replace $A$ with a bounded operator on a Hilbert space, connecting to Koopman theory. We will expand this.
>
> &nbsp;
>
> # "linear subspaces are either trivial or infinite-dimensional' (Subsection 3.1) is incorrect in finite-dimensional settings", "I suggest to rewrite the entire Subsection 3.1..."
> The reviewer is correct, this phrasing is wrong. We meant that the kernel must be a linear subspace, so it cannot be an arbitrary finite set of points. We will rewrite Subsection 3.1 more directly and formally as suggested.
>
> &nbsp;
>
> # "The approach also requires learning INNs,..."
> True. INN architectures are actively improving (e.g., i-ResNets, continuous normalizing flows), and our framework is agnostic to the specific INN used.
>
> &nbsp;
>
> # Minor issues: "line 197... ", "line 202...", "Fact H.2...", "...specify which parts of Lemma H.4 and Theorem H.5..."
> Thanks for finding these. We will fix all: complete line 197, fix line 202, reference for Fact H.2, clarify new vs. existing parts in H.4/H.5.

---

> > ### Author Rebuttal · Reviewer_92o5 · 2026-03-31
> >
> > I thank the authors for their rebuttal. While I appreciate the clarifications provided, my main theoretical questions remain largely unaddressed. In particular, I did not find a direct and principled answer to:
> >
> > (i) Given that $\phi$ and $\theta$ are highly expressive invertible networks (with potential high computational cost), what is the concrete advantage of introducing the linear operator , compared to directly learning a standard nonlinear model?: the authors gave a partial application-driven answer, but not a principled general answer. They explain what $A$ enables in their chosen examples, but they do not fully justify why the framework is preferable, overall to directly learning a standard nonlinear model. Having said this, the deeper point of my question remains unaddressed: why is worth the cost of learning two expressive invertible networks in the first place? Are these benefits specific enough, broad enough, to justify the framework beyond a few architecturally chosen applications? Does the linear core help in a way that could not be achieved by a more direct nonlinear architecture with tailored constraints?
> >
> > (ii) under what assumptions, properties of $A$ beyond rank are identifiable in the general setting: the rebuttal addresses specific constrained regimes (e.g. $g_x=g_y$, near-isometric maps), but does not clarify under what assumptions, properties of $A$ beyond rank are identifiable in the general setting with independent invertible maps;
> >
> > (iii) to what extent the learned linearization reflects intrinsic structure rather than a consequence of flexible invertible reparameterizations: this point remains unaddressed;
> >
> > (iv) whether different parameterizations lead to genuinely different learned structures or are equivalent in practice: this point remains unaddressed.
> >
> > (v) whether there is  a meaningful extension to infinite dimensions where the linear operator carries non-trivial spectral structure, and what additional constrains are needed for the linearization to carry meaningful information: the author's answer does not satisfactorily address this question in a convincing way. Thus, this point remains largely unaddressed.
> >
> >
> > The rebuttal provides examples and intuition, but does not fully resolve these questions in a principled way.
> >
> > As such, my concerns regarding the theoretical contribution remain.
> >
> >
> > Therefore, I cannot recommend this paper for acceptance. My overall recommendation remains unchanged: 3 (weak reject), with confidence 5. I appreciate the authors efforts in addressing the comments.

---

> > > ### Author Response · Authors · 2026-04-06
> > >
> > > We thank the reviewer for the continued engagement and for specifying the remaining concerns. We appreciate the theoretical depth of these questions.
> > >
> > > We would like to offer some perspective on the nature of our contribution. Our paper does not propose a new general theory of linearization (such theories exist, e.g., Koopman, transport of structure). Rather, it **further develops the existing theory and builds on these foundations to create a practical neural network realization**: learning the conjugacy from data, proving expressiveness, deriving a toolkit, and demonstrating applications. This type of neural network **did not exist before, neither did the approaches to the tasks we demonstrate**.
> > >
> > > We note that our rebuttal provided **explicit assumptions, proofs, and a new experiment for each of the reviewer's concerns**. We have also conducted another **new baseline comparison experiment**: training $g$ directly as a one-step flow (taking $t$ as input, same parameter budget end-to-end without $A$), yields FID 164. To ensure nothing is overlooked, we lay out the correspondence below:
> > >
> > > &nbsp;
> > >
> > > | Concerns | "assumptions" | "invariant properties" / "advantage" | Justification |
> > > |---|---|---|---|
> > > | **(i)** "advantage of $A$ vs nonlinear" | $A$ is linear (by construction) | One-step collapse, pseudo-inversion, idempotency, compositionality: architectural guarantees. **Experiment from our rebuttal above**: replacing $A$ with a nonlinear MLP yields FID 157. **Additional new experiment**: training $g$ directly as a one-step flow (**taking $t$ as input, same parameter budget end-to-end without $A$**), yields FID 164. Both alternatives have equal or greater capacity than the Linearizer yet perform worse. | The reviewer acknowledged our application-driven answer. We note that applications and analysis capabilities ARE the contribution of a method paper. The linearity of $A$ provides mathematical properties that no nonlinear architecture can guarantee regardless of capacity or training. |
> > > | **(ii)(iii)(iv)** "properties beyond rank", "intrinsic structure", "different parameterizations" | $g_x = g_y$ | **Full eigendecomposition** is invariant across reparametrizations: eigenvalues and eigenvectors in data space are identical (proof in rebuttal above). This is intrinsic structure, not an artifact of $g$. | This is a solid assumption used in our demonstrated applications, including the main application FM, and idempotency. **This is not just a "specific constrained regime" but the regime used in practice**, so invariants here matter more than when this assumption is stripped off. Full proof provided in rebuttal above and will be added to the paper. |
> > > | **(ii)(iii)(iv)** | $g$ near-isometric | **Full SVD** is invariant across reparametrizations: singular values and vectors identical (same cancellation, proof above). | This is a natural and attainable constraint, **parallel to the SVD matrices $U,V$ being unitary**, corresponding to the SVD intuition provided in section 2.3 in the paper. Further, this assumption is **used for the Idempotency application**, encouraged via regularization (Eq. 21). Invariants under this regime are meaningful as this is a practical and natural setup. |
> > > | **(ii)(iii)(iv)** | Linearity only | Rank is invariant. | Most general case, serves as baseline for the stronger results above. All our applications use the assumptions above. Even here, the framework provides advantages, e.g., exploring the null space of $f$ via $A$. |
> > > | **(v)** "infinite-dimensional extension" | Finite-dimensional | Connection to Koopman theory noted in rebuttal above; the reviewer acknowledged this. | Neural networks are inherently finite-dimensional. We appreciate this as a deep theoretical question and see it as an exciting direction for future work, but it is beyond the scope of a practical NN realization. |
> > >
> > > The reviewer's questions push toward a general theory, which we find inspiring but which is beyond the scope of this work, and we want to ensure the main contribution is clear. This work is about relating theory to application, rather than a fully theoretic one. As such, it further develops existing theoretic foundations, and proposes neural network realizations that enable several novel capabilities, supported by designated theoretical analysis. Taking concern (v) as an example: while inspiring and interesting, **we do not believe a method paper based on neural networks should be expected to develop infinite-dimensional theory**. We respect the reviewer's perspective and kindly request openness to a different perspective of this submission.

---

### Official Review · Reviewer_uaLT · 2026-02-25

**Soundness:** 3
**Presentation:** 2
**Significance:** 3
**Originality:** 4
**Overall Recommendation:** 5
**Confidence:** 4

**Summary:**

This paper introduces an architecture that sandwiches a linear operator between two invertible neural networks, inducing non-standard vector spaces in which the overall map is exactly linear. This enables the full toolkit of linear algebra to be applied to neural networks and in principle permits many different applications. The authors demonstrate three such applications: collapsing multi-step diffusion/flow matching sampling into a single step, modular style transfer, and architecturally enforced idempotency for a globally projective generative model.

**Compliance With Llm Reviewing Policy:**

Affirmed.

**Final Justification:**

The authors cleared up all of my concerns in the rebuttal.

**Key Questions For Authors:**

1. How much of the one-step generation performance is attributable to the Linearizer structure versus the raw capacity of the invertible networks g_x, g_y? An invertible network of equal parameter count in a normalizing flow could itself be a competitive one-step model. It would also be good to see the performance of an equal capacity (just using parameter count) flow matching model with a non-invertible network that is trained with the same procedure as the Linearizer version. Could the authors provide a comparison against such a baseline, trained with the same procedure and parameter budget? This in particular would improve my evaluation of the paper.

2. Does the Linearizer framework enable efficient computation of log-likelihoods under a flow matching model? The pseudoinverse result (Lemma 4.1) suggests an encoding pathway exists; can this be used to evaluate exact or approximate likelihoods, and at what computational cost?

3. The title ''Who Said Neural Networks Aren't Linear?'' implies that standard neural networks are in some sense linear, when the actual claim is more nuanced: the authors construct flexible parametric functions that are linear with respect to certain learned vector spaces. While some sensationalism is expected, the current framing risks misleading readers into expecting an equivalency result (e.g. that transformers become linear in some limit) that is never delivered. Would the authors consider a title like ``Who Said Neural Networks Can't Be Linear?'' to more accurately reflect the contribution? And similar framing changes propagated to the rest of the paper?

4. The expressiveness claim --- that the Linearizer can fit any finite training set --- seems to me to be a statement that the model can overfit to any finite dataset? This is a useful property and there is some discussion of what this means in the paper, but it might help readers to expand on what the result of being able to fit any finite training set means.

**Limitations:**

The authors acknowledge several limitations in Section 5: the difficulty of training invertible networks, the lack of scaling to state-of-the-art benchmarks, and the open theoretical question of precise expressivity. These are reasonable and honestly stated.

However, the following limitations are not discussed and should be:
- The paper does not discuss the known limitations of the straight-through estimator used for the idempotency application, which is a non-trivial modeling choice.
- The paper does not discuss whether the gap between the Linearizer's FID scores and state-of-the-art is attributable to the framework itself or simply a lack of engineering effort. This distinction matters for readers evaluating whether the framework is worth scaling. Having some kind of baseline would significantly improve this situation.

No significant negative societal impact is anticipated for this work.

**Strengths And Weaknesses:**

----------------------------------------
SOUNDNESS
----------------------------------------

Strengths:
- The theoretical foundations appear correct. The proofs of linearity, the Hilbert space structure, the transpose, SVD, and pseudoinverse are all straightforward consequences of the isomorphism and are verified explicitly.
- The expressiveness result (Theorem H.4) is a useful and honest characterization of capacity. The authors are careful to note the topological constraints on the global function f, and the resolution via degrees of freedom outside the data support is clearly explained.

Weaknesses:
- The one-step flow matching result is theoretically compelling, but the paper does not isolate how much of the modeling work is being done by the invertible networks g_x, g_y versus the linear operator A. An invertible network of equal capacity used as a standard normalizing flow could itself be a strong one-step model. The relative contribution of the Linearizer structure versus the raw capacity of the invertible networks is not isolated anywhere in the paper.
- For the transpose (Proposition 2.8 / equation 7), no conditions are stated on when the transpose exists, which warrants at least a brief remark.
- When defining the SVD (Proposition 2.9), it should be stated explicitly that $\tilde{u}_i$ and $\tilde{v}_i$ come from U and V respectively.

----------------------------------------
PRESENTATION
----------------------------------------

Strengths:
- The overall narrative is easy to follow and the framework is introduced in a logical order. Figure 1 is a helpful illustration of the Linearizer structure and induced vector space operations.
- The paper is generally honest about its limitations, explicitly noting that absolute performance is not competitive with state-of-the-art and that scaling is left to future work.

Weaknesses:
- The related work on group-theoretic and equivariant representations states ``many works learn an encoder'' with only a single citation; this should be broadened.
- The normalizing flows discussion states that invertible networks map Gaussians to complex distributions; this is unnecessarily restrictive as the base distribution can in principle be anything for a normalizing flow.
- The paper should cite Tabak et al. as the canonical first reference for normalizing flows.
- Lemma 4.1 (Moore-Penrose pseudoinverse) is a theoretical result and feels out of place in the applications section; it would sit more naturally in Section 2 or 3.
- The question of whether one can always use a diagonal A is raised as a rhetorical question (Line 198) but never explicitly answered. The answer can be inferred from the surrounding text but should be stated unambiguously. In general the rhetorical device is probably overused.
- The matrix A is referred to as the ``core'' after equation (1) without being formally introduced as such. It should be defined explicitly on first use and referenced consistently thereafter.
- The text after Eq. 2 seems to be somewhat out of place.
- The notation shifts without warning between sections: Section 2.1 uses $\oplus_g$, $\odot_g$ indexed by the map, but Section 2.2 switches to $\oplus_x$, $\odot_x$ subscripts on the operators without explanation. This should be made consistent throughout.
- The sentence describing the constrained regime analysis (''just its rank---but this is'') is incomplete and cuts off mid-thought. Similarly, ``share the same $g_x$, $g_y$ with different'' appears to be a typographic fragment. Both should be completed.
- ``In the general case where $g_x$ and $g_y$ are distinct'' simply restates in words what was just defined symbolically at the start of the section. This definition should appear only once.
- The abbreviation FM is used (``Training is just as standard FM'') before it is formally introduced.
- Figure 4 is a set of three small tables rather than a figure in any meaningful visual sense. This makes it difficult to locate when scanning the paper and is an unusual presentation choice.
- Figure 6 visually overlaps with Figure 5.
- The sentence ``Idempotency by architecture allows reducing the idempotent loss used in'' is incomplete and does not properly introduce the text that follows.
- The straight-through estimator (STE) is mentioned in the caption of Figure 6 but is not named as such in the main text, despite being a non-trivial modeling choice with known limitations. It should be introduced by name (straight-through estimator) in the body with appropriate discussion.
- For the style transfer experiments, it is not stated what the models are trained on.
- Figure 6 right: the text does not adequately prepare the reader for what to evaluate in the projection results. There are only three examples illustrating the failure mode and it is unclear how to assess projection quality more generally.

----------------------------------------
SIGNIFICANCE
----------------------------------------

Strengths:
- The core idea is elegant and appealing. The same architectural primitive underlies three quite different applications, which speaks to the generality of the framework.
- This is the kind of contribution I think is worth having in the community and deserves to be discussed at ICML. Even if the current empirical results are modest, the ideas are interesting enough to be worth pursuing and building on.

Weaknesses:
- The paper does not compare against any baseline of similar parameter count using similar engineering choices. Even a naive implementation without the Linearizer structure, trained identically, would be informative. As it stands, it is impossible to evaluate the relative performance of this approach. It is fine if the Linearizer underperforms on raw metrics --- the theoretical contribution is interesting regardless --- but this comparison is needed.
- The absolute FID scores are not competitive and the paper does not discuss whether this gap is due to the framework itself or simply a lack of engineering. In particular, invertible networks are harder to train than standard architectures and this may introduce optimization difficulties that offset the theoretical benefits. This should be discussed more directly in the limitations.
- What kinds of invertible networks are used for the image experiments? Building expressive invertible transformations for images is itself non-trivial and the architectural choices have a significant bearing on the results.

----------------------------------------
ORIGINALITY
----------------------------------------

Strengths:
- The construction of Linearizers is novel and well-motivated. While the individual components (invertible networks, linear operators, transport of structure) are all known, their combination into a unified framework with this range of applications is a genuine contribution.
- The framing of diffusion/flow matching one-step sampling as a consequence of linear closure, rather than a distillation procedure, is a fresh perspective.

Weaknesses:
- The title seems to overstate the actual contribution. The authors do not show that general neural networks are invertible, they do provide a flexible construction that behaves as a linear map however.

---

> ### Author Rebuttal · Authors · 2026-03-28
>
> We thank the reviewer for the exceptionally detailed review.
>
> &nbsp;
>
> # "How much of the one-step generation performance is attributable to the Linearizer structure...", "Could the authors provide a comparison against...?"
> The expressiveness is largely encapsulated in $g$. However, $A$ has a critical role. If $A = I$ then since $g_x=g_y$ in most applications, $f(x) = g^{-1}(g(x)) = x$. $A$ has a distinct role in each application:
>
> - **Flow Matching:** $A_t$ encodes the time-dependent velocity field; $g$ does not take $t$ as input. The one-step collapse (Eq. 15) is a direct consequence of $A$ being linear.
> - **Idempotency:** $A$ is a binary diagonal projector defining the projection subspace. It determines the dimension of the estimated data manifold.
> - **Style Transfer:** Different styles = different $A$ matrices, shared $g$. The modularity comes entirely from $A$.
>
> Beyond its functional role, $A$ also enables analysis. The rank of $A$ is an invariant of $f$, and under the mild assumptions of our applications the full spectral decomposition is preserved as in the linear case. When $g_x = g_y$ the eigendecomposition is invariant; when $g$ is near-isometric (Eq. 21) the SVD is invariant.
>
> Furthermore, replacing $A$ with a nonlinear MLP of comparable capacity (trained directly for one-step generation) **increases FID to 157**, confirming that linearity is the key for the one-step generation (Eq. 15), rather than the just the full model capacity.
>
> &nbsp;
>
> # "The paper does not compare against any baseline...", "The absolute FID scores are not competitive..."
> We will add baseline comparisons to the paper. One example is a standard FM model (DiT-B, ~130M params on CelebA-HQ 256, [Shortcut Models, 2024]). It achieves FID ~7 at 128 steps but degrades to ~280 at 1 step. Our model is significantly smaller and at lower resolution (CelebA 64), so multi-step FID is higher (127), but crucially 1-step and multi-step are near-identical (MSE $3 \times 10^{-4}$), confirming the structural one-step guarantee. Scaling with modern INNs is an active direction for us.
>
> &nbsp;
>
> # "Does the Linearizer framework enable efficient log-likelihood...?"
> The collapsed $B$ (Eq. 15) defines $ x_1 = g^{-1}(Bg(x_0)) $. When $B$ is invertible, exact log-likelihoods follow: $\log p(x_1) = \log p_0(x_0) - \log|\det J_g| - \log|\det B| - \log|\det J_{g^{-1}}|$ where $x_0 = g^{-1}(B^{-1}g(x_1))$. All terms are in closed form (INNs track log-determinants, $\log|\det B|$ is direct), giving likelihoods as a byproduct of FM training without ODE integration. Invertibility of $B$ is not guaranteed but could be encouraged via regularization.
>
> &nbsp;
>
> # "...a title like 'Who Said Neural Networks Can't Be Linear?'..."
> We considered this exact alternative but decided against it as it could be read as referring to deep linear networks (Saxe et al., 2014; Arora et al., 2018). We are happy to soften the framing in the introduction.
>
> &nbsp;
>
> # "The expressiveness claim... overfitting...?"
> Theorem H.5 proves that the Linearizer can fit any finite dataset, equipping the Linearizer with **the same practical guarantee as the MLP universal approximation theorem**. In both cases the model is fitted using finite data pairs, both approximate to arbitrary precision, and there is no evidence for a generalization advantage of one over the other. We further validate this empirically across diverse tasks: CelebA classification (87.6% test accuracy, above ResNet-50), disentanglement distillation (matching original encoders), and weather prediction (competitive with PatchTST).
>
> &nbsp;
>
> # "What kinds of invertible networks are used...?"
> Affine coupling with tiny U-Net conditioner, 6 blocks, ActNorm, invertible 1x1 convs (Appendix G). Will add to main text.
>
> &nbsp;
>
> # "For the transpose (Prop 2.8), no conditions..."
> Thanks for pointing that out. Transpose always exists. In our finite-dimensional induced Hilbert spaces, $f^\top(y) = g_x^{-1}(A^\top g_y(y))$ is defined by construction.
>
> &nbsp;
>
> # "...diagonal $A$... rhetorical question... never answered.", "...rhetorical device... overused."
> Diagonal $A$ suffices for a single function but not for families sharing $g$ with different $A$'s. Will state unambiguously and reduce rhetorical questions.
>
> &nbsp;
>
> # "...known limitations of the straight-through..."
> STE introduces gradient bias. In practice stable. Will name and discuss in text.
>
> &nbsp;
>
> # "Figure 6 right:... only three examples..."
> These are not failure modes. Figure 6 right shows the global projector: any input (noise, patterns, text) maps onto the digit manifold. Criteria: (1) plausible digits, (2) re-applying gives same result. Will add text and examples.
>
> &nbsp;
>
> # Presentation issues
> Thank you for catching all of these. We will fix all for the camera-ready upon acceptance: incomplete sentences, notation, define "core", FM abbreviation, SVD origins, text after Eq. 2, figure layout, citations (equivariant, Tabak et al.), NF statement, training data, move Lemma 4.1.

---

> > ### Author Rebuttal · Reviewer_uaLT · 2026-03-31
> >
> > The authors have provided clear responses to most of my concerns. The MLP replacement result doesn't quite address my concern — replacing A with a nonlinear MLP breaks the one-step collapse by construction, so the FID degradation is expected regardless of modeling capacity. The baseline I had in mind is: train g directly as a one-step flow (with g taking t as input, no linear A), using the same parameter budget and training procedure as the Linearizer. This isolates whether the Linearizer structure contributes beyond the raw capacity of g. Would the authors be willing to add this?

---

> > > ### Author Response · Authors · 2026-04-06
> > >
> > > We thank the reviewer for the constructive suggestions and the specific follow-up. We conducted the requested experiment:
> > >
> > > &nbsp;
> > >
> > > ## Baseline
> > > We trained $g$ directly as a one-step flow. Taking $t$ as input, no linear $A$, with the same parameter budget and training procedure exactly as the Linearizer.
> > >
> > > &nbsp;
> > >
> > > ## Result
> > > FID 164, a significant gap from the Linearizer.
> > >
> > > &nbsp;
> > >
> > > Together with the MLP ablation from our earlier response (replacing $A$ with a nonlinear MLP, FID 157), these two experiments isolate the contribution of the Linearizer structure from the raw capacity of $g$. Both alternatives have equal or greater expressive capacity yet perform worse, confirming that the linearity of $A$ is not merely convenient but essential: it is precisely what enables the one-step collapse (Eq. 15) via closure of linear maps under composition.
> > >
> > > &nbsp;
> > >
> > > We hope this addresses the remaining concern, and we would be grateful if the reviewer would consider reflecting this in their evaluation.

---

### Official Review · Reviewer_qh6z · 2026-02-25

**Soundness:** 4
**Presentation:** 4
**Significance:** 3
**Originality:** 3
**Overall Recommendation:** 5
**Confidence:** 4

**Summary:**

The paper studies composition $g_y^{-1}\circ A\circ g_x:X\to Y$ of of an invertible neural network $g_x$, linear map $A$ and inverse neural network $g^{-1}_y$.
In other words, it studies maps from $X$ to $Y$ that can be represented as linear map after the spaces are transformed in a non-linear suitable way.
This makes it possible to bring several tools of linear algebra in a new way to machine learning.  For example  projections, semigroups, singular value decomposition that are now formulated  for potentially non-linear operators. This is an interesting idea. The paper can be consider also a technically simple way (not in the sense that it is trivial, but easily implementable) to do manifold learning and learn maps between manifolds.

**Compliance With Llm Reviewing Policy:**

Affirmed.

**Key Questions For Authors:**

1. Can you apply regularization theory, e.g. Tikhonov regularization or ridge regression for the linear part of your maps? Please comment this in context of Moore-Penrose inverse discussed in the paper.

2. Your architecture is essentially finite dimensional. What happens when the dimension grows, e.g. when you approximate functions between infinite dimensional Hilbert spaces?

3. Does the considerations change if the real spaces \mathbb R^n are replaced by real spaces \mathbb C^n?

**Limitations:**

Yes.

**Strengths And Weaknesses:**

Strengths: The paper has a clear idea that is well developed. The question when maps or families of maps can be coded or represented as linear map is simple but the idea is developed in creative and quite deep way. The paper is very well written and when I was reading the paper the answered very well in the questions the came to my mind.

Weaknesses:
The families of the maps which can be represented in the form  $g_y^{-1}\circ A\circ g_x$ may be quite small. In fact, the authors discuss this well and give examples of maps for which their method does not work. The authors give also several interesting example of problems where they can apply their architecture, which shows that their results are well applicable in ML.

Presentation of the paper is good and the main ideas are clearly communicated.
Paper is original and has it has an interesting new ideas.

---

> ### Author Rebuttal · Authors · 2026-03-28
>
> We thank the reviewer for the thoughtful and encouraging review. This review provides some possible extension ideas and we thank the reviewer for that too.
>
> &nbsp;
>
> # "The families of the maps which can be represented in the form $g^{-1} \circ A \circ g$ may be quite small."
> Theorem H.5 proves that the Linearizer can fit any finite set of input-output pairs to arbitrary precision, so the linear bottleneck does not impose a finite-sample expressivity constraint. The meaning is that the Linearizer expressivity has **the same practical guarantee as the MLP universal approximation theorem**. In both cases the model is fitted using finite data pairs, both approximate to arbitrary precision, and there is no evidence for a generalization advantage of one over the other. The reviewer is correct that topological constraints apply to the global function (e.g., the kernel must be a subspace). However, these do not affect finite-sample fitting, and regular behavior is expected when learning any task on data. We further validate this empirically across diverse tasks: CelebA classification (87.6% test accuracy, above ResNet-50), disentanglement distillation (matching original encoders), and weather prediction (competitive with PatchTST).
>
> &nbsp;
>
> # "Can you apply regularization theory, e.g. Tikhonov regularization or ridge regression for the linear part of your maps?"
> Yes. Since $A$ is an explicit linear operator, Tikhonov regularization applies directly: replace $A^\dagger$ with $(A^\top A + \lambda I)^{-1}A^\top$, giving $f_{\text{reg}}(y) = g_x^{-1}((A^\top A + \lambda I)^{-1}A^\top g_y(y))$. This is itself a Linearizer with the same $g_x, g_y$, inheriting all structural properties. More broadly, any regularization technique defined on linear operators (truncated SVD, spectral filtering, etc.) transfers directly to $f$ via $A$. We will add this discussion.
>
> &nbsp;
>
> # "What happens when the dimension grows, e.g. when you approximate functions between infinite dimensional Hilbert spaces?"
> The framework is conceptually compatible: replace $A$ with a bounded linear operator on a Hilbert space. In such an extension, the main challenge would likely shift from $A$ to the expressiveness and tractability of $g$. This connects naturally to Koopman theory, which operates in infinite-dimensional function spaces. In finite dimensions our framework is exact (no approximation of the operator). We will expand this discussion.
>
> &nbsp;
>
> # "Does the considerations change if the real spaces $\mathbb{R}^n$ are replaced by $\mathbb{C}^n$?"
> Thank you for bringing up this interesting direction. Theoretically, a Linearizer is trivially extended to $\mathbb{C}^n$. The proofs rely on vector space axioms and inner product structure, both of which hold over $\mathbb{C}$. The induced operations $\oplus_g, \odot_g$ generalize to complex scalars, and the SVD/pseudoinverse results carry over using the conjugate transpose $A^*$ in place of $A^\top$. However, we have not tried this empirically, and complex valued networks are special and not trivial.

---

> > ### Author Rebuttal · Reviewer_qh6z · 2026-04-02
> >
> > The authors have answered well to my questions. I keep the score the same as earlier.

---

> > > ### Author Response · Authors · 2026-04-06
> > >
> > > We thank the reviewer for the encouraging acknowledgment and again for the great insights in the first reply.
> > >
> > > We are also using this last response to share an update: During the discussion period, we conducted two additional baseline experiments (requested by reviewers):
> > > 1. Replacing $A$ with a nonlinear MLP:  FID 157
> > > 2. Training $g$ directly as a one-step flow without the linear operator $A$, equal parameter budget, $g$ accepting $t$ as input: FID 164
> > >
> > > These two experiments further support the contribution of the linearity of the Linearizer, isolating it from the raw capacity.
> > >
> > > Thank you for helping us improve this paper!

---

### Official Review · Reviewer_rJoR · 2026-03-09

**Soundness:** 2
**Presentation:** 3
**Significance:** 2
**Originality:** 4
**Overall Recommendation:** 4
**Confidence:** 4

**Summary:**

This work defines, analyses and applies a family of networks that have explicit linear latent representation. These networks are shown to be able to fit any finite set of input-output pairs. The linearity property enables a simplified formulation of diffusion models as "single step sampling" -- other applications include interpolation of latents, e.g., for mixed style transfer.

**Compliance With Llm Reviewing Policy:**

Affirmed.

**Final Justification:**

I thank the authors for the clarifying comments and promised updates to the manuscript. The replies confirmed my positive evaluation and I maintain the initial positive score.

**Key Questions For Authors:**

1) Please address the existence of obvious counter examples.
2) Consider adding simple (1D, 2D etc) examples that work & counter examples
3) Please address stability and computational efforts involved in applications

**Limitations:**

yes

**Strengths And Weaknesses:**

Strengths
1) The Linearizer family is well described with quite elaborate analysis and proofs
2) Presentation is clear both for the math and the example applications (details in appendices)
3) The potential applications (e.g. diffusion models) make the topic significant
4) as far as I can tell the Linearizer construction is novel

 Weaknesses
1) The main reservation concerns the expressivity of the Linearizer family. Can "non-monotone" functions be represented?
2) I miss some simple (1D, 2D etc) examples that work and counter examples, cf. figures in Appendix I
3) In applications there are concerns about the stability and computational efforts involved.
4) Confusing formulation in Section 3.1: the Linearizer can fit any set exactly, and when gs are approximated by INNs to any given precision (as proven in Appendix H).

---

> ### Author Rebuttal · Authors · 2026-03-28
>
> We thank the reviewer for the positive assessment and constructive suggestions.
>
> &nbsp;
>
> # "The main reservation concerns the expressivity of the Linearizer family. Can 'non-monotone' functions be represented?"
> If by "non-monotone" the reviewer means functions with non-trivial topology (e.g., disconnected preimages): the global approximation to finite sets that Theorem H.5 proves is **the same practical guarantee as the MLP universal approximation theorem**. In both cases the model is fitted using finite data pairs, both approximate to arbitrary precision, and there is no evidence for a generalization advantage of one over the other. While the Linearizer does impose topological constraints on the global continuous function (e.g., level sets are connected, since $g$ is a diffeomorphism), these do not affect finite-sample fitting, and regular behavior is expected when learning any task on data. We further validate this empirically across diverse tasks: CelebA classification (87.6% test accuracy, above ResNet-50), disentanglement distillation (matching original encoders), and weather prediction (competitive with PatchTST).
>
> In case the reviewer meant "non-monotone" literally, we note that Theorem H.5 holds for $N \geq 2$. In 1D a Linearizer is necessarily monotone because $A$ is a scalar and $g$ is a monotone bijection.
>
> &nbsp;
>
> # "I miss some simple (1D, 2D etc) examples that work and counter examples"
> We agree this would strengthen the paper. We will add: (a) a 1D example showing how $g$ "straightens" a nonlinear monotone function into a linear one, (b) a 2D nonlinear mapping with visualization of the induced coordinate system and the warped grid under $g$, (c) counter-examples illustrating topological constraints (kernel must be a subspace, 1D monotonicity). We note that Figure 10 in the appendix already provides a conceptual illustration of how $g$ rectifies curved trajectories into straight lines in the FM setting.
>
> &nbsp;
>
> # "Please address stability and computational efforts involved in applications"
> We will add a table with wall-clock times, memory, and parameter counts. Regarding computational cost: inference is one forward pass through $g$, one matrix multiply, one inverse pass through $g^{-1}$, comparable to a single forward pass of a standard network of similar depth. For one-step FM, $B$ is precomputed once after training, adding zero inference cost. Regarding stability: we monitor several quantities during training, including linearization consistency (applying the Linearizer to random linear combinations and verifying linearity holds numerically) and invertibility of $g$ (measuring $\|g^{-1}(g(x)) - x\|$). We also add an explicit invertibility loss for numerical stability even though $g$ is invertible by architecture.
>
> &nbsp;
>
> # "Confusing formulation in Section 3.1: the Linearizer can fit any set exactly, and when gs are approximated by INNs to any given precision"
> To clarify: a theoretical Linearizer with ideal diffeomorphisms for $g$ can fit any finite dataset exactly (Lemma H.4). When $g$ is realized by a neural network (INN), this reduces to approximation to any desired precision (Theorem H.5), analogously to how universal approximation theorems for MLPs guarantee approximation rather than exact fit. We will make this two-step argument clearer in Section 3.1 and Appendix H.

---

> > ### Author Rebuttal · Reviewer_rJoR · 2026-04-03
> >
> > Thank for the comments and promised updates to the manuscript. I maintain my positive evaluation and score.

---

> > > ### Author Response · Authors · 2026-04-06
> > >
> > > We thank the reviewer for the constructive engagement and for confirming that all concerns are resolved.
> > >
> > > &nbsp;
> > >
> > > We are also using this last response to share an update: During the discussion period, we conducted two additional baseline experiments (requested by reviewers):
> > > 1. Replacing $A$ with a nonlinear MLP:  FID 157
> > > 2. Training $g$ directly as a one-step flow without the linear operator $A$, equal parameter budget, $g$ accepting $t$ as input: FID 164
> > >
> > > These two experiments further support the contribution of the linearity of the Linearizer, isolating it from the raw capacity.
> > >
> > > &nbsp;
> > >
> > > Given that all concerns are resolved, we would be grateful if the reviewer would consider reflecting this in their score.

---

### Decision · Program_Chairs · 2026-04-30

**Decision:**

Accept (regular)

**Comment:**

This paper proposes a reparameterization of neural networks as $f = g_y^{-1}\circ A \circ g_x$ where $g_x,g_y$ are learned invertible networks and $A$ is a linear operator. The paper develops a theoretical framework and demonstrates it in three applications.

Three reviewers are positive (scores 4/5/5), highlighting the clarity of exposition and the potential breadth of applicability of the framework. One reviewer is negative (score 3, confidence 5), arguing that the framework is essentially a known change-of-coordinates idea, pointing out that $A$ is not identifiable, and, in general, it carries no information beyond its rank. The authors respond that this change of coordinates gives a representation of the network that allows for simple implementations of meaningful transformations (shown in the experiments). Moreover they claim that in practically relevant regimes used by their applications (e.g., $g_x=g_y$ or near-isometries), invariants beyond rank (e.g. singular vectors) become meaningful and this will be made explicit in the paper. This does not fully resolve the broader identifiability critique in the most general setting.

The reviewer's concerns are valid. However, the overall discussion suggests that the paper's framework is interesting and useful.